# CUDAFORGE: AN AGENT FRAMEWORK WITH HARDWARE FEEDBACK FOR CUDA KERNEL OPTIMIZATION

## ABSTRACT

Developing efficient CUDA kernels is increasingly critical for AI applications such as large-scale LLM training. However, manual kernel design is both costly and time-consuming, motivating automatic approaches that leverage LLMs for code generation. Existing methods for automatic kernel generation, however, often produce low-efficiency kernels, incur high computational overhead, and fail to generalize across settings.

In this work, we propose `CudaForge`, a training-free multi-agent workflow for CUDA kernel generation and optimization. Our workflow is inspired by the iterative workflow of human experts, which contains steps such as developing initial kernels, testing correctness, analyzing hardware feedback, and iterative improvement. More specifically, `CudaForge` employs two LLM agents – a Coder and a Judge – that iteratively generate, correct, and optimize CUDA kernels, while integrating hardware feedback such as Nsight Compute (NCU) metrics. In our extensive evaluations, we show that `CudaForge` , by leveraging base models like OpenAI-o3, achieves 97.6% correctness of generated kernels and an average 1.68× speedup over PyTorch baselines, substantially surpassing state-of-the-art models including OpenAI-o3 and Kevin on KernelBench. Beyond accuracy and speed, `CudaForge` demonstrates strong generalization across GPUs (A100, RTX 6000, 4090, 3090) and base models (OpenAI-o3, GPT5, gpt-oss-120B, Claude-Sonnet-4, QwQ-32B), while maintaining high efficiency. In particular, generating an optimized kernel takes about 25 minutes on one RTX 6000 and incurs $0.30 API cost. Our results highlight that multi-agent, training-free workflows can enable cost-effective, generalizable, and high-performance CUDA kernel optimization.

## 1 INTRODUCTION

**Motivation.** CUDA has become the *de facto* standard for deep learning training because modern frameworks such as PyTorch and TensorFlow are deeply integrated with NVIDIA's optimized GPU libraries (NVIDIA, 2025b). Efficient CUDA kernels are crucial for accelerating deep learning workloads(Dao et al., 2022; Dao, 2024) .

However, developing high-efficiency cuda kernels has been known as challenging with very high learning curve, requiring deep expertise in GPU architectures and parallel programming(Li et al., 2024). For example, it took more than 2 years from the debut of the Hopper GPU architecture to the release of FlashAttentionV3 (Shah et al., 2024), which is specially designed for Hopper GPUs.

This high development barrier has driven growing interest in finding automated ways of generating highly efficient and customized CUDA kernels. For example, some work (Tillet et al., 2019) (Chen et al., 2018) employs auto-tuning and evolutionary search to automatically explore kernel implementation spaces and optimize low-level parameters for specific hardware. More recently, there has been a growing interest in leveraging large language models (LLMs) to perform such tasks. LLM is believed to hold great promise in generating efficient and high-quality kernels, due to its capability of code generation in other domains, such as Python, C++ and Triton (Dong et al., 2025; Jiang et al., 2024; Anonymous, 2025; Li et al., 2025b; Woo et al., 2025; Li et al., 2025a).

**Existing Works and Key Challenges.** Generally, using LLMs for CUDA kernel generation is still in an early stage. In KernelBench (Ouyang et al., 2025), the authors attempt to directly use state-of-

the-art (SOTA) models, such as OpenAI-o1 and Claude-3.5-Sonnet, to generate kernels. However, it has been observed that these SOTA models still struggle to produce correct or performant kernels out of the box, revealing fundamental limitations of existing LLMs in this domain.

To address this gap, recent studies have explored two main paradigms. The first approach is based on reinforcement learning (RL) (Schulman et al., 2017; Shao et al., 2024). CUDA-L1 (Team, 2025) and Kevin (Baronio et al., 2025) adopt RL to enhance LLMs' ability to generate correct and performant CUDA code. The second approach is based on AI agents. In particular, in an independent and contemporaneous work (Lange et al., 2025)[1], researchers have explored agentic frameworks at inference time. Agents project PyTorch method into CUDA kernel design, then the CUDA kernels are further refined by sampling new kernels and verification filtering. This design effectively improves correctness in CUDA kernel generation without the high cost of RL training.

Despite these advances, several key challenges remain:

**(C1) Limited kernel efficiency.** While RL-based methods improve LLMs' ability to generate CUDA kernels, their optimization capability remains insufficient. For example, Kevin-32B only achieves an average speedup of $1.10\times$ over KernelBench test cases, even after sampling 16 parallel trajectories with 8 refinement turns each per kernel (Baronio et al., 2025). Further, CUDA-L1 often fails to directly optimize the CUDA kernels, but producing official implementation of PyTorch (Team, 2025) (see Appendix F for details).

**(C2) High training and inference cost.** RL-based approaches such as (Team, 2025; Baronio et al., 2025) require substantial computational resources and long training cycles, making them unsuitable for low-resource or rapid-prototyping settings. In addition, multi-stage agentic pipeline developed by (Lange et al., 2025) incurs high inference costs (about 6 H100 hours and $5 API cost per kernel), which greatly limits its practical applicability of the approach.

**(C3) Lack of hardware feedback.** Human experts typically follow an iterative workflow to develop performant CUDA kernels through testing and refinement. They rely on hardware feedback like Nsight Compute (NCU)[2] to identify bottlenecks and optimize kernels accordingly (Wu et al., 2025; NVIDIA, 2025a; Hu et al., 2025). In contrast, RL-based approaches (Team, 2025; Baronio et al., 2025) train LLMs to directly generate or optimize kernels, but do not incorporate hardware feedback at all. As a result, they rely on blind exploration during generation, lacking the targeted guidance. This often leads to suboptimal kernel efficiency, limiting their practical applicability.

These challenges raise a natural question: *Can we design a simple but effective hardware-aware approach that reliably produces efficient CUDA kernels at low cost?*

**Our Contributions.** To address these challenges, we propose `CudaForge`, **a simple, effective and low-cost** multi-agent workflow for CUDA kernel generation and optimization, as shown in Figure 1. Our workflow is inspired by the iterative workflow of human experts (Wu et al., 2025; NVIDIA, 2025a; Hu et al., 2025), which contains steps such as developing initial kernels, testing correctness, analyzing hardware feedback, and iterative improvement.

This workflow involves two specialized LLM agents that iteratively generate and optimize CUDA kernels: a Coder, which generates kernels given task instructions and Judge feedback, and a Judge, which analyzes kernels and hardware feedback to guide the Coder generation. One key novelty of `CudaForge` is its integration of external hardware feedback, including GPU specifications and Nsight Compute (NCU) metrics, enabling the Judge to identify performance bottlenecks like human experts and provide targeted optimization guidance to the Coder.

Compared to single-LLM approaches that generate and evaluate code using the same LLM, our framework separates these roles into an *independent* Coder and Judge, enabling more specialized reasoning and more reliable iterative refinement. Unlike RL-based methods, `CudaForge` is training-free, avoiding the substantial cost of policy training. It is also hardware-aware, allowing it to tailor CUDA kernel optimizations to the underlying system, making the proposed framework easily generalizable across different GPUs. Finally, in contrast to existing multi-agent frameworks (Lange et al., 2025), `CudaForge` is lightweight and cost-efficient, running in just 25 minutes on a single RTX6000 GPU and $0.3 per kernel in API costs, while still achieving better performance.

---

[1]published on arxiv Sept 16th, 2025
[2]Nsight Compute (NCU) is NVIDIA's official kernel-level profiler for CUDA programs.

We evaluate `CudaForge` on 250 KernelBench tasks from Level 1 to Level 3. Though these tasks are challenging, `CudaForge` attains a 97.6% correctness rate and delivers an average speedup of 1.68× over PyTorch baselines, which significantly outperforms advanced RL model like Kevin-32B and advanced frontier model like OpenAI-o3 (OpenAI, 2025). Further, we have conducted comprehensive ablation studies of the features of `CudaForge`, such as its effectiveness across multiple GPU architectures, its inference-time scalability by increasing the number of generation, and the effect of different base models. Overall, we observed that the proposed `CudaForge` achieves robust performance in all these settings.

These findings highlight the key contribution of this work: The proposed LLM agent workflow `CudaForge` is simple but effective: at very low cost, it develops performant CUDA kernels for many practical tasks, for a variety of GPU architectures and base models. It also exhibits strong test-time scaling capabilities where solution quality can improve substantially while increasing its iteration rounds. These results demonstrate `CudaForge`'s strong practical applicability.

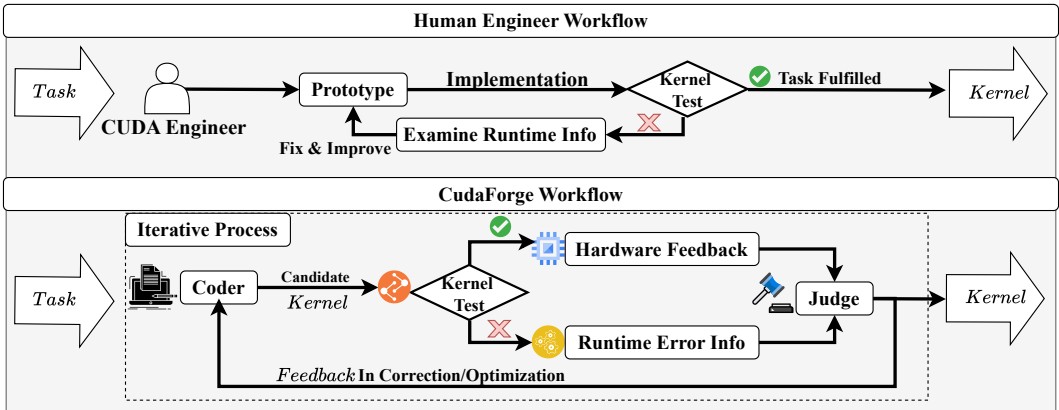

Figure 1: Comparison between human and `CudaForge` workflows. Top: Human experts iteratively refine kernels by writing a prototype, testing it, and analyzing runtime feedback. Bottom: `CudaForge` mimics human workflow with two specialized agents (Coder and Judge). The Coder generates candidate kernels, while the Judge analyzes runtime info and hardware feedback to provide correction or optimization feedback. The process iterates until it reaches maximum round $N$.

## 2 THE CUDAFORGE FRAMEWORK FOR CUDA KERNEL OPTIMIZATION

### 2.1 CUDAFORGE FRAMEWORK

Given a CUDA kernel generation task, the objective is to generate a kernel that is functionally equivalent to its PyTorch reference while achieving the lowest possible execution latency.

Inspired by the iterative workflow of human experts (Wu et al., 2025; NVIDIA, 2025a; Hu et al., 2025), we design `CudaForge` as an iterative multi-agent framework, illustrated in Figure 1. The framework involves two independent agents: a **Coder** and a **Judge**. The Coder generates candidate kernels based on the task description and feedback from the Judge, while the Judge evaluates each candidate using the kernel itself, hardware feedback and runtime information.

More specifically, given a CUDA kernel generation task, the Coder first receives the task requirements and PyTorch reference implementation, then produces an initial candidate kernel. This candidate is compiled and executed on test cases to check correctness. If it fails, the Judge inspects *runtime information* (e.g., compilation errors, mismatched outputs with the PyTorch reference) and analyzes the faulty kernel. It then returns correction feedback (e.g., missing header file) to guide the next iteration. Once a kernel candidate passes the correctness test, the Judge profiles it with the NCU tool to obtain NCU metrics (e.g., memory throughput, occupancy, warp efficiency). Together with GPU specifications, these metrics form the *hardware feedback* that allows the Judge to identify the dominant bottleneck (e.g., compute-bound or memory-bound) and provide one specific optimization feedback (e.g. using shared memory) to the Coder.

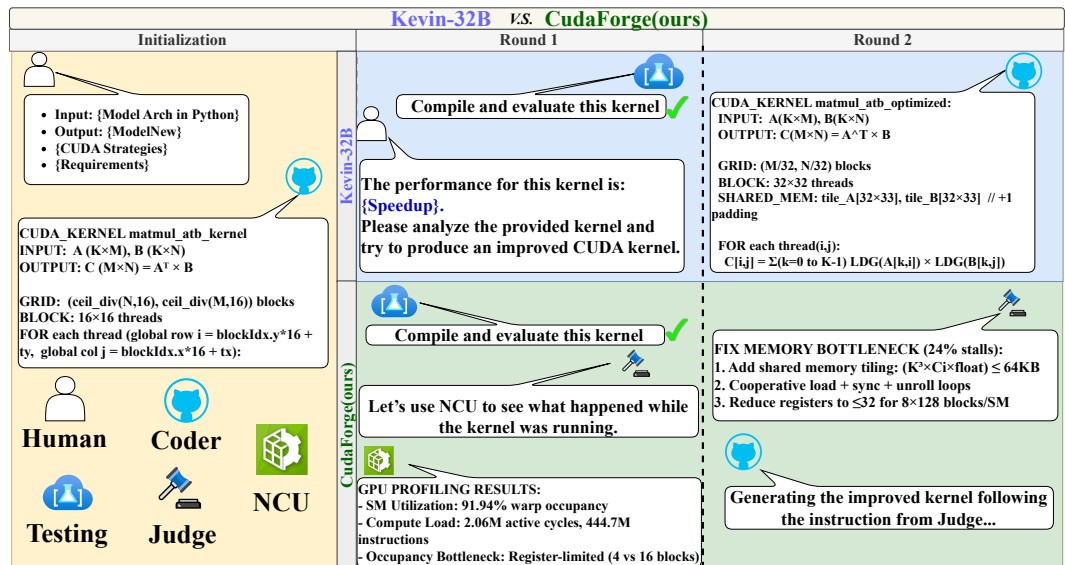

Figure 2: The overview of conversation between agents in Kevin-32B and CudaForge.

In the next iteration, the Coder is prompted with the previous kernel, Judge feedback, and the original task requirements, and generates a corrected or optimized kernel. This process repeats for up to $N$ iterations, after which we select the most efficient correct kernel as the final solution.

CudaForge achieves reliability and efficiency through **three key design choices**. First, it adopts a two-agent system where the Coder focuses on generation and the Judge on evaluation, separating the "cognitive" load (See Section 3.4). The Coder receives only feedback from the Judge, while the Judge uses hardware and runtime information to guide generation and optimization. This division of labor mirrors human workflows and mitigate the risk of overlooking errors or inefficiencies. Second, the framework follows an iterative optimization process, progressively correcting errors and improving efficiency across rounds. This enables stable refinement, especially on hard tasks. Third, it explicitly incorporates hardware feedback, such as GPU specifications and NCU metrics, so the Judge can pinpoint bottlenecks and provide actionable guidance to the Coder. This targeted optimization avoids blind exploration and ensures directed performance gains.

## 2.2 HOW TO INTEGRATE HARDWARE FEEDBACK

In this subsection, we describe in detail a key design consideration, which enables CudaForge to utilize hardware feedback for kernel performance optimization. The hardware feedback module integrates static GPU specifications (e.g. architecture, memory bandwidth, per-thread register limits, per-SM shared-memory capacity) with performance metrics (e.g. memory throughput, occupancy, and warp efficiency) from Nsight Compute (NCU) collected during kernel execution. By cross-referencing GPU specifications and NCU metrics, the Judge infers the kernel's primary performance-limiting cause and bottleneck mechanism. Figure 2 illustrates how Judge uses the hardware feedback to optimize kernels.

Just as CUDA engineers focus on key indicators, we do not pass the entire set of NCU metrics to the Judge. Feeding all metrics can overwhelm the decision process with excessive, partially redundant signals and lead to unstable judgments (See Appendix E.1 for detail). Instead, we design a novel protocol which profiles a subset of critical metrics provided by NCU and forward them to Judge so that we can improve the quality of the judge outputs. More specifically, the key subset of metrics are selected off-line (before the agent start to work), through the following steps:

**(Step 1)** Kernel sampling and Selection: We first profile key metrics on some preselected representative tasks (e.g., Conv2D, MatMul) to prepare a reliable metric set. Specifically, for each task we run 100 self-refine (repeating the cycle generating → execute/profile → evaluate → repair/optimize) with a single SOTA model (e.g., OpenAI-o3), collect the generated and correct kernels, and select 10 with the largest speed disparity (fastest vs. slowest). See Algorithm 1.

---

**Algorithm 1** Step 1: Kernel Sampling and Selection

---

**Input:** Task set $Task = \{T_1, T_2, \ldots, T_n\}$
**Output:** Selected subsets $K_i^*$ for each task $T_i$
**for** $i \leftarrow 1$ **to** $n$ **do**
    $K_i \leftarrow \emptyset$ **for** $j \leftarrow 1$ **to** $100$ **do**
        $k_j \leftarrow$ generate_kernel$(T_i)$   $K_i \leftarrow K_i \cup \{k_j\}$
    **end**
    Sort $K_i$ in nondecreasing order according to kernel runtime   $m \leftarrow |K_i|$ ;   // Here $m = 100$
    $K_i^* \leftarrow \{K_i[1], K_i[2], K_i[3], K_i[4], K_i[5], K_i[m-4], K_i[m-3], K_i[m-2], K_i[m-1], K_i[m]\}$
**end**

---

**(Step 2)** Top-20 metrics within each task: We then refine the metrics within each task to identify the most relevant candidates. Specifically, for each task we consolidate the NCU metrics from the 10 kernels selected from Step 1 into a single dataset. Since Nsight Compute reports a consistent full set of metrics across all kernels, the metric categories are aligned by default. We then remove aliases and strongly collinear indicators, and compute Pearson correlations between each metric and kernel runtime. We retain only the Top-20 metrics (by absolute correlation) as the candidate set for that task (see Appendix E.2 for examples).

**(Step 3)** Metrics selection across-tasks: Finally, we consolidate metrics across tasks to build a stable, task-agnostic set. We compare the Top-20 lists across tasks and keep metrics that consistently appear, show the same correlation direction, and achieve high global scores. This yields 24 metrics that are strongly correlated with kernel runtime across tasks. See Algorithm 2.

---

**Algorithm 2** Step 2-3: Profiling and Metrics Selection

---

**Input:** $K^* = \{K_1^*, K_2^*, \ldots, K_n^*\}$, where each $K_i^* = \{k_1^*, k_2^*, \ldots, k_{10}^*\}$
**Output:** Final metrics set $Final\_Metrics$ containing 24 unique metrics
$M^* \leftarrow \emptyset$
**for** $i \leftarrow 1$ **to** $n$ **do**
    $M_i^* \leftarrow \emptyset$ **foreach** $k \in K_i^*$ **do**
        $M \leftarrow$ NCU_Profile$(k)$ ;          // Run NCU profiling, $M = \{m_1, m_2, \ldots, m_j\}$
        **foreach** $m \in M$ **do**
            Compute Pearson correlation coefficient $r(m, \text{runtime}(k))$
        **end**
        $Top20(k) \leftarrow$ the 20 metrics in $M$ with highest $|r(\cdot, \text{runtime}(k))|$   $M_i^* \leftarrow M_i^* \cup Top20(k)$
    **end**
    $M^* \leftarrow M^* \cup M_i^*$
**end**
// Final set contains 24 distinct metrics
$Final\_Metrics \leftarrow \bigcap_{i=1}^{n} M_i^*$   $|Final\_Metrics| = 24$

---

After the above steps are completed offline, during kernel optimization, the Judge profiles each generated kernel with NCU and uses only this 24 metrics as references (see Appendix E.3 for details).

Overall, at each iteration, the Judge collects hardware feedback, including static GPU specifications and the key subset of NCU metrics. Based on this information, the Judge identifies the dominant bottleneck by analyzing the 24 metrics and runtime log. To prevent AI agent searching without direction and generating random results, the Judge only captures 3-4 most important metrics in each round according to its own reasoning. For example, Judge can identify the current kernel is memory-bound when memory throughput is high but computing resources utilization is low, and then it will choose memory related metrics as critical metrics in this round. After this, Judge will generate suggestions on how to modify the kernel to address current critical bottleneck. The Coder incorporates this guidance in the next round generation accordingly. This mechanism enables our multi-agents system focus on addressing only one critical program bottleneck in each round, eventually optimize overall kernel performance step by step in iterative rounds, just like human expert's real workflow.

# 3 EXPERIMENTS

## 3.1 BENCHMARK AND EVALUATION

We evaluate our method on **KernelBench** (Ouyang et al., 2025), a popular benchmark designed to assess the ability of LLMs to generate CUDA kernels. KernelBench consists of multiple difficulty levels; we adopt all tasks from Level 1 to Level 3, resulting in a total of 250 tasks. Specifically, Level 1 contains relatively simple 100 tasks involving basic operators (e.g., matrix multiplication), Level 2 includes medium-difficulty 100 tasks composed of multi-step operator combinations, and Level 3 contains 50 challenging tasks involving full neural network architectures (e.g., AlexNet). Each task is accompanied by a reference PyTorch implementation and predefined input/output specifications, which enables fully automated and reliable evaluation of both correctness and performance.

We evaluate model performance on KernelBench using the following metrics:

(1) **Correctness**: the fraction of tasks for which the generated kernel compiles successfully and produces outputs identical to the PyTorch reference on all test cases. (2) **Performance**: the ratio of the execution speed (tested on a specific GPU), between a correct generated kernel and its PyTorch reference. (3) $\mathbf{fast}_p$: the proportion of correct kernels whose execution speed exceeds $p\times$ that of the PyTorch reference (e.g., $\text{fast}_1$ indicates faster than PyTorch). (4) **Median speedup**: the median of 'Performance' values across all tasks, reflecting typical rather than average behavior. (5) **75th percentile speedup**: the 75th percentile of Performance values, capturing upper-quartile efficiency.

For methods that perform iterative refinement or generate multiple candidates (including `CudaForge`), we report the best-performing correct kernel among all candidates for each task. De­tails of test cases, correctness evaluation and performance evaluation could be found in Appendix B.

## 3.2 SETTINGS & BASELINES

In our main results, we instantiate `CudaForge` with OpenAI-o3 as both the Coder and the Judge as our *default* setting. We set the maximum number of iteration rounds to $N{=}10$ to balance perfor­mance improvements and inference cost. Unless otherwise stated, all methods are evaluated under the same compilation/runtime environment in Quadro RTX 6000 and task-specific test suites.

To contextualize the performance of `CudaForge` and assess the effect of advanced foundation mod­els, we include the following baselines for main results and ablation studies: (1) O3-S: OpenAI-o3 (single-shot), one-pass generation without iteration; (2) O3-10: OpenAI-o3-10-round (self-refine), ten rounds of self-refinement without a Judge, where the model relies solely on itself to correct and optimize kernels given hardware feedback; (3) O3-10-C: OpenAI-o3-10-round (correction­only), a variant of `CudaForge` where the Judge provides only correctness feedback but no per­formance optimization feedback; (4) O3-10-O: OpenAI-o3-10-round (optimization-only), a variant of `CudaForge` where the Judge provides only optimization feedback but no correction feedback; (5) Kevin-10: Kevin-32B-10-round(self-refine), the RL-based model run for ten iterative rounds under the same protocol; (6) AgentBaseline: the agentic workflow from (Lange et al., 2025), a strong multi-agent baseline. Due to the high computational cost of running Kevin-32B on the full benchmark, we additionally construct a stratified random subset $\mathcal{D}^*$ for fair comparison. Details of KernelBench and $\mathcal{D}^*$ are provided in Appendix G.

This suite enables a comprehensive comparison across (i) base model vs. corresponding agent-based method, (ii) the presence/absence of Judge feedback, (iii) RL-based vs. training-free agent-based approaches, and (iv) different agentic methods.

## 3.3 MAIN RESULTS

Table 1 reports the main results on KernelBench. `CudaForge` consistently outperforms all base­lines across all metrics, both on the full benchmark $\mathcal{D}$ and on the stratified subset $\mathcal{D}^*$ .

On $\mathcal{D}$, `CudaForge` attains **97.6%** correctness with an average performance of $\mathbf{1.677\times}$, and **70.8%** $\mathbf{Fast}_1$, while achieving a median speedup of $1.107\times$ with a 75th percentile speedup of $1.592\times$. This is a clear improvement over its base model O3-S. On $\mathcal{D}^*$, which allows fair comparison with the advanced RL model Kevin, `CudaForge` achieves 100% correctness, a median speedup of $1.322\times$,

Table 1: Main results on KernelBench (Level 1-3, 250 tasks). Results of AgentBaseline is on Level 1 and 2. All experiments here are run in RTX 6000. Methods evaluated on $\mathcal{D}^*$ are marked with $*$.

| Method | Correct↑ | Median ↑ | 75% ↑ | Perf ↑ | Fast$_1$↑ |
|---|---|---|---|---|---|
| O3-S | 57.6% | 0.390 | 1.014 | 0.680 | 31.60% |
| O3-10 | 90.8% | 1.012 | 1.209 | 1.107 | 55.20% |
| O3-10-C | 97.6% | 1.031 | 1.238 | 1.222 | 59.60% |
| O3-10-O | 88.4% | 1.061 | 1.483 | 1.509 | 64.00% |
| AgentBaseline | 95.0% | — | — | 1.490 | — |
| Kevin-10$^*$ | 64.0% | 0.472 | 1.047 | 0.608 | 36.00% |
| CudaForge | **97.6%** | **1.107** | **1.592** | **1.677** | **70.80%** |
| CudaForge$^*$ | **100%** | **1.322** | **1.736** | **1.767** | **84.00%** |

Table 2: Main results on KernelBench (Level 1-3, 250 tasks) of CudaForge.

| Task | Correct↑ | Median ↑ | 75% ↑ | Perf ↑ | Fast$_1$↑ |
|---|---|---|---|---|---|
| Level 1 | 96% | 1.044 | 1.751 | 1.448 | 54.0% |
| Level 2 | 100% | 1.124 | 1.427 | 2.104 | 89.0% |
| Level 3 | 96% | 1.081 | 1.510 | 1.283 | 68.0% |

a 75th percentile speedup of $1.736\times$, an average performance of $1.767\times$, and 84.0% Fast$_1$. This substantially surpasses Kevin-10, which reaches only 64.0% correctness, $0.472\times$ median, $1.047\times$ at the 75th percentile, $0.608\times$ performance, and 36.0% Fast$_1$. This represents a +63.6% absolute gain in correctness and a $+1.159\times$ speedup, despite CudaForge being a training-free method while Kevin is a RL-trained model.

We also compare CudaForge with AgenticBaseline in KernelBench Level 1 and Level 2[3]. As shown in Table 2, CudaForge achieves 98% correctness and an average speedup of $1.776\times$, which outperforms AgenticBaseline (95.0%, $1.490\times$), especially in speedup. This result shows our advantage compared to existing agentic work.

Notably, on Level 3—the most challenging tier of KernelBench—CudaForge achieves **96%** correctness and an average **$1.283\times$** speedup. Given the complexity of Level 3 tasks, which involve full neural network architectures and multi-stage operations, these results demonstrate that CudaForge is capable of reliably generating and optimizing highly complex CUDA kernels, where prior approaches (Baronio et al., 2025; Lange et al., 2025) have not explored it.

We evaluate both API and time cost on KernelBench. On average, CudaForge requires only **25 minutes** on a single RTX6000 GPU and incurs **\$0.3** API cost per kernel. This is highly cost-efficient compared with another agentic work (Lange et al., 2025), which reports about **6 GPU hours on H100** and **\$5** per kernel in their Appendix E. These results demonstrate that, by leveraging hardware feedback, our workflow can rapidly converge to high-quality solutions at low cost. Details of where the 25 minutes is spent could be found in Appendix C.

## 3.4 Ablation Studies

**Comparison with O3-10 (self-refinement).** A key motivation behind CudaForge is to decouple the roles of generation and evaluation. In O3-10, the same model performs ten rounds of self-refinement, implicitly taking on both roles: it must both propose new kernels and evaluate its own outputs based on hardware feedback and runtime signals. While this strategy raises correctness 57.6% to 92.8%, performance remains limited ($1.107\times$ speedup, 55.2% Fast$_1$). In contrast, CudaForge explicitly separates responsibilities: the Coder focuses on code generation, while the Judge specializes in providing structured feedback. This division of labor proves critical—allowing each agent to concentrate on a distinct reasoning process—and results in significantly higher efficiency ($1.677\times$ speedup, 70.8% Fast$_1$) without sacrificing correctness.

---

[3]Note that these works only report results in Level 1 and 2, and we directly take the results from their paper since the paper has not opened sourced the code.

| Method | Correct↑ | Performance↑ | $Fast_1$↑ |
|---|---|---|---|
| CudaForge-Top 24 metrics(ours) | 100% | 1.767 | 84% |
| CudaForge-Full metrics | 100% | 1.414 | 80% |
| CudaForge-Random 24 metrics | 100% | 1.655 | 76% |
| CudaForge-Top 5 metrics(ours) | 100% | 1.641 | 76% |
| CudaForge-Top 10 metrics(ours) | 100% | 1.644 | 80% |
| CudaForge-Top 20 metrics(ours) | 100% | 1.827 | 88% |

Table 3: Ablation study on NCU metric selection. Comparing full, random, and top-$k$ subsets shows that concise and carefully chosen metrics (Top-24) provide strong overall performance, while Top-20 offers slightly higher speedup with similar behavior.

**Comparison with O3-10-C (correction-only Judge).** In O3-10-C, the Judge only provides correction feedback based on runtime signals, without optimization feedback. This setting achieves the same 97.6% correctness as `CudaForge`, confirming that iterative error correction is sufficient to ensure reliable kernel generation. However, efficiency remains much lower, with only $1.222\times$ performance and 58.8% $Fast_1$. The contrast with `CudaForge`(**$1.677\times$**, **70.8%**) highlights that while correctness feedback stabilizes generation, performance feedback—grounded in hardware profiling—is essential for driving substantial efficiency gains.

**Comparison with O3-10-O (optimization-only Judge).** We also evaluate the variant O3-10-O, where the Judge provides only optimization feedback, without correction feedback. In this setting, the Coder frequently generates kernels that fail to compile or run, since functional errors remain uncorrected. As a result, this setting achieves 88.4% correctness and a $1.509\times$ speedup, which are substantially lower than `CudaForge`($1.677\times$, 70.8%). The result demonstrates that correction feedback plays a significant role in `CudaForge`'s performance. The absence of it will lead to lower correctness. And without first ensuring functional validity, optimization feedback alone is ineffective and often wasted.

**Ablation study on NCU metrics.** A key design choice in CudaForge is to filter the full set of NCU metrics and retain a subset of 24 critical metrics for the Judge. This selective design enables the Judge to focus on the most informative performance indicators while avoiding redundancy and inconsistent feedback. To evaluate this choice, we conduct an ablation study comparing our top-24 metric subset against several variants, including using all NCU metrics, using a random subset of 24 metrics, and using smaller subsets of the top-5, top-10, and top-20 metrics.

As shown in Table 3, two conclusions emerge. First, using the complete set of NCU metrics degrades both correctness and speedup, as the Judge becomes overwhelmed by excessive and partially redundant profiler signals. Second, selecting too few metrics or selecting them randomly restricts the Judge's ability to provide meaningful optimization feedback, resulting in inferior performance. Our 24-metric design consistently achieves the best overall results across all variants, while the top-20 subset yields slightly higher performance but remains very close in practice.

Furthermore, profiling with all NCU metrics significantly increases computational and API cost: each kernel requires approximately 40 minutes on an RTX 6000 GPU and incurs roughly $1 in API usage. In contrast, our selective design reduces runtime to about 25 minutes and API cost to $0.3 while achieving superior performance. These findings demonstrate that concise, carefully curated hardware feedback is both more effective and more efficient than exhaustive profiling. We further provide a case study illustrating this phenomenon in Appendix E.1.

## 3.5 GENERALIZATION CAPABILITY OF CUDAFORGE

In this section, we analyze `CudaForge`'s capabilities across various maximum iteration num $N$, GPU architectures and base models. Considering the high cost of full experiment, we use the stratified subset $\mathcal{D}^*$ for this section.

**Scaling up the maximum number of iteration rounds** We investigate the effect of the maximum iteration number $N$ on `CudaForge`'s performance.

Table 4: `CudaForge`'s performance on different GPUs. The system consistently achieves high correctness and strong performance across architectures by incorporating GPU specifications and *Nsight Compute* profiling signals during optimization.

| GPU | Correct↑ | Median ↑ | 75% ↑ | Perf ↑ | Fast$_1$↑ |
|---|---|---|---|---|---|
| RTX 6000(Ada Arch-Data center level) | 100% | 1.322 | 1.736 | 1.767 | 84.0% |
| RTX 4090(Ada Arch-Desktop level) | 100% | 1.188 | 1.589 | 1.327 | 80.0% |
| A100(Ampere Arch-Data center level) | 100% | 1.371 | 1.762 | 1.841 | 84.0% |
| RTX 3090(Ampere Arch-Desktop level) | 100% | 1.155 | 1.706 | 1.320 | 72.0% |

Table 5: Performance of `CudaForge` with different base model combinations. We fix one agent as OpenAI-o3 and replace the other with various models. All combinations achieve strong results, showing that the framework is not tied to a specific base model.

| Models (Coder/Judge) | Correct↑ | Median ↑ | 75% ↑ | Perf ↑ | Fast$_1$↑ |
|---|---|---|---|---|---|
| O3 / O3 | 100% | 1.322 | 1.736 | 1.767 | 84.0% |
| O3 / GPT-5 | 100% | 1.131 | 1.561 | 2.114 | 96.0% |
| O3 / Claude | 100% | 1.265 | 1.456 | 1.829 | 84.0% |
| O3 / GPT-OSS-120B | 100% | 1.226 | 1.490 | 1.364 | 76.0% |
| GPT-5 / O3 | 100% | 1.125 | 1.388 | 1.896 | 72.0% |
| Claude / O3 | 88% | 1.052 | 1.207 | 1.398 | 56.0% |
| GPT-OSS-120B / O3 | 96% | 1.080 | 1.477 | 1.653 | 68.0% |
| QwQ / O3 | 84% | 0.965 | 1.153 | 0.790 | 44.0% |

As shown in Figure 3, increasing $N$ from 1 to 10 leads to substantial performance gains, indicating that `CudaForge`can rapidly improve kernel efficiency through iterative refinement. Further increasing $N$ from 10 to 30 continues to improve performance, though with a slower growth rate, suggesting that the system gradually approaches its performance ceiling. These results demonstrate that `CudaForge` benefits from test-time scaling and has the potential to achieve even stronger performance given larger $N$ with additional inference cost.

**Using `CudaForge` in different GPUs.** We also evaluate `CudaForge` on various GPU architectures, including RTX 6000, RTX 4090, RTX 3090 and A100, to examine its effectiveness under different hardware conditions. As shown in Table 4, `CudaForge` consistently achieves high correctness and strong performance on all tested GPUs. This is a direct consequence of its design: during the optimization phase, the Judge explicitly incorporates hardware feedback, including NCU metrics and GPU specifications when generating feedback to Coder. This allows the Coder to produce kernels that are tailored to the target GPU at inference time, without training.

**Instantiate `CudaForge` with various LLM.** To examine whether `CudaForge` depends on a specific base model, we conduct experiments by fixing one side (Coder or Judge) as *OpenAI-o3* and replacing the other with various advanced LLMs, including *QwQ-32B*, *GPT-5*, *Claude*, and *GPT-OSS-120B*. As shown in Table 5, all combinations achieve high correctness and strong performance,

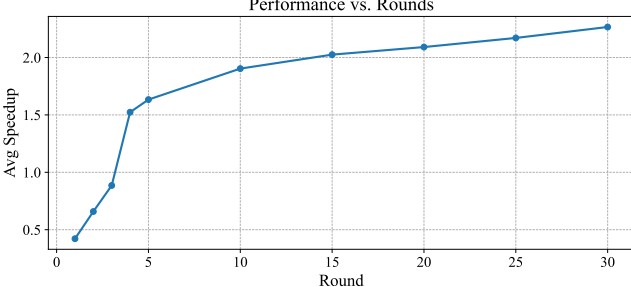

Figure 3: Scaling the number of iteration rounds to 30 on KernelBench (subset $\mathcal{D}^*$).

comparable to or even surpassing the original O3/O3 configuration. These results indicate that `CudaForge` is not tied to a specific base model: its effectiveness stems from the workflow of Coder and Judge, and it can readily benefit from stronger models as they emerge.

# 4 SUPPLEMENT EXPERIMENTS AND OBSERVATIONS

**Case study.** To comprehensively understand the details of `CudaForge`, we investigate to a specific case to study its iterative workflow. As shown in Appendix A., it demonstrate a 10-round refine process of KernelBench Level 1 task 95. Our workflow iteratively corrects and optimizes the kernel, with the feedback of Judge model. More details could be found in A.

**Observations in CUDA-L1 results.** We carefully examined the kernel outputs reported by CUDA-L1 (see Appendix F) and identified an interesting phenomenon that we term *"fake kernels."* These kernels, while reported as performant, often contain no actual CUDA code. Instead, they rely on `try-except` constructs and fall back to PyTorch's official implementations to solve the task. This observation highlights a fundamental challenge in evaluating LLM-generated CUDA kernels. To avoid this issue, we have manually checked all kernels in our experiments.

# 5 CONCLUSION

We presented `CudaForge`, a training-free multi-agent framework for CUDA kernel generation and optimization. The framework mimics the iterative workflow of human experts, explicitly incorporating hardware feedback to guide targeted kernel refinement rather than blind exploration. On the KernelBench benchmark, `CudaForge` achieves highest correctness rate and significant performance gains compared with all existing method, while also demonstrating robustness across diverse GPU architectures and base LLMs Moreover, its performance scales effectively with the number of refinement rounds. Finally, thanks to its low API and time cost, `CudaForge` provides a practical and efficient solution for automated CUDA kernel development.

ETHICS STATEMENT

This paper proposes the `CudaForge` framework for automatically generating and optimizing CUDA kernels, applied to diverse tasks in KernelBench. The design and experiments strictly adhere to ethical guidelines, ensuring that no sensitive or personally identifiable information is involved. All experiments rely solely on publicly available benchmarks and standard GPU hardware, and no human data was collected or processed.

We acknowledge the potential environmental concerns related to large-scale model training. While our framework reduces inference-time cost compared to RL-based methods, more efficient kernels could indirectly accelerate resource-intensive workloads. We therefore encourage responsible and sustainable use of this technology. Our framework is intended strictly for research and scientific purposes, and does not introduce additional risks beyond those already associated with standard compiler or optimization tools.

REPRODUCIBILITY STATEMENT

We have taken extensive measures to ensure the reproducibility of our work. All experiments are conducted on the publicly available KernelBench benchmark, which provides standardized tasks, PyTorch references, and input/output specifications. We report detailed results across all difficulty levels, including averaged metrics and stratified subsets, to ensure statistical robustness.

To support replication, we provide a comprehensive description of our workflow in Section 2.1 and include all prompts used for the Coder and Judge agents in Appendix D. Furthermore, we provide full experimental details, including GPU hardware platforms, evaluation metrics, and iteration protocols. Since `CudaForge` is entirely training-free, no additional data collection or model training is required, greatly simplifying reproducibility.

We will release code and experiment scripts upon publication, ensuring that all results reported in this paper can be faithfully reproduced.

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

# A  CASE STUDY

## A.1  A GOOD CASE

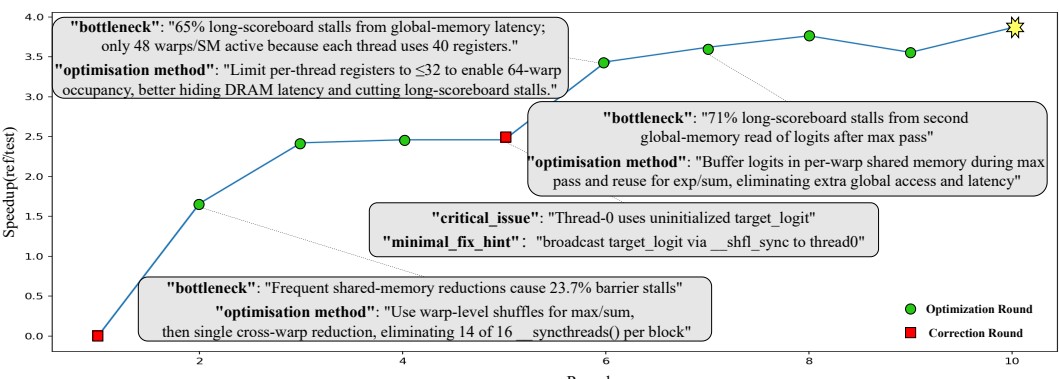

Figure 4: Illustration of the Judge's outputs—bottleneck diagnoses and optimization suggestions—on KernelBench Level-1 Task 95 (CrossEntropyLoss), as well as the correspondi speedup across rounds (green = optimization, red = correction).

In this section, we present a case study on a single task to illustrate how the Judge diagnoses issues and recommends optimizations. Figure 4 depicts the 10-round refinement process of CudaForge on task 95_CrossEntropyLoss. We highlight four representative rounds—three optimization rounds and one repair round—to demonstrate how the Judge leverages hardware feedback from NCU to provide targeted optimization or bug-fix suggestions.

In round 2, which is an optimization round, the Judge notices that 23.7% of active warps are stalled due to barrier-type dependencies, which means roughly one quarter of potential issue opportunities are blocked by synchronization. According to this, the Judge recommended replacing the original shared-memory reduction that required multiple block-level synchronizations with a warp-level shuffle reduction, giving below suggestion as prompt for coder: use warp-level shuffles in the max and sum phases, then perform a single cross-warp combine, reducing __syncthreads() per block from 16 to 2 (a reduction of 14). After applying this change, performance improved from $1.66\times$ to $2.42\times$, with barrier stalls reduced and instruction-issue efficiency increased.

In round 5, it is a correction round. The previous round fails a numerical check with the following error: "Outputs are not close, indicating a result mismatch". The Judge diagnosed the root cause as an uninitialized target_logit in thread 0 ("Thread-0 uses uninitialized target_logit"), which means the variable target_logit is not updated to thread 0, leading wrong computing results. Accordingly, the Judge gave the minimal fix suggestion, broadcast target_logit via _shfl_sync to thread 0. After applying the fix, the numerical issue disappeared.

In Rounds 6 & 7 (both optimization rounds), the Judge continues to track smsp__warp_issue_stalled_long_scoreboard_per_warp_active.pct. In Round 6, this metric is about 65%, primarily reflecting long-scoreboard stalls caused by global-memory latency. Per-thread register usage is high, resulting in limited occupancy (only $\sim 48$ active warps/SM) and insufficient latency hiding. The recommendation is to reduce per-thread registers to raise concurrency to $\sim 64$ warps/SM and thereby lower the long-scoreboard share. In Round 7, the metric rises to about 71%, rooted in a second global read of logits after the max pass. The Judge therefore advises buffering logits in per-warp shared memory during the max pass and reusing them in the expsum phase, eliminating the redundant global memory access. Together, these strategies reduce global memory access, significantly cut long-scoreboard stalls, improve issue efficiency and throughput; after these two rounds, the speedup increases from $3.436\times$ to $3.762\times$.

This task demonstrates our CudaForge's stability and expert-like workflow: first analyzing bottlenecks from hardware feedback, then deriving the corresponding optimization strategy.

## A.2   A BAD CASE

We also explore some bad cases of our workflow. Take KernelBench Level 1 Task 5 as an example:

```python
1   import torch
2   import torch.nn as nn
3
4   class Model(nn.Module):
5       """
6       Simple model that performs a matrix-scalar multiplication (C = A * s)
7       """
8       def __init__(self):
9           super(Model, self).__init__()
10
11      def forward(self, A: torch.Tensor, s: float) -> torch.Tensor:
12          """
13          Performs matrix-scalar multiplication.
14
15          Args:
16              A: Input matrix of shape (M, N)
17              s: Scalar value
18
19          Returns:
20              C: Resulting matrix of shape (M, N)
21          """
22          return A * s
23
24  M = 16384
25  N = 4096
26
27  def get_inputs():
28      A = torch.randn(M, N)
29      s = 3.14
30      return [A, s]
31
32  def get_init_inputs():
33      return []  # No special initialization inputs needed
```

In the first round, `CudaForge` generated a wrong kernel:

```python
1   ```python
2   # <complete ModelNew code>
3   import torch
4   import torch.nn as nn
5   from torch.utils.cpp_extension import load_inline
6
7   # ----------------------------------------
8   # CUDA source for fused ReLU + per-channel bias addition
9   # ----------------------------------------
10  cuda_src = r"""
11  #include <torch/extension.h>
12  #include <cuda.h>
13  #include <cuda_runtime.h>
14
15  template <typename scalar_t>
16  __global__ void relu_bias_add_kernel(const scalar_t* __restrict__ x,
17                           const scalar_t* __restrict__ bias,
18                           scalar_t* __restrict__ y,
```

```
19                        const int spatial,
20                        const int channels,
21                        const int total_elems) {
22      int idx = blockIdx.x * blockDim.x + threadIdx.x;
23      if (idx >= total_elems) return;
24
25      int c = (idx / spatial) % channels;   // channel index
26      scalar_t v = x[idx];
27      v = v > static_cast<scalar_t>(0) ? v : static_cast<scalar_t>(0);
28      v += bias[c];
29      y[idx] = v;
30    }
31
32    torch::Tensor relu_bias_add_cuda(torch::Tensor x, torch::Tensor bias) {
33      TORCH_CHECK(x.is_cuda(), "Input tensor x must reside on CUDA device");
34      TORCH_CHECK(bias.is_cuda(), "Bias tensor must reside on CUDA device");
35      TORCH_CHECK(x.scalar_type() == torch::kFloat32,
36              "Only float32 data type is supported");
37
38      const int N = x.size(0);
39      const int C = x.size(1);
40      const int H = x.size(2);
41      const int W = x.size(3);
42      const int spatial = H * W;
43      const int total_elems = N * C * spatial;
44
45      auto y = torch::empty_like(x);
46
47      const int threads = 256;
48      const int blocks  = (total_elems + threads - 1) / threads;
49
50      relu_bias_add_kernel<<<blocks, threads>>>(
51          x.data_ptr<float>(),
52          bias.view({C}).data_ptr<float>(),
53          y.data_ptr<float>(),
54          spatial,
55          C,
56          total_elems);
57
58      cudaError_t err = cudaGetLastError();
59      if (err != cudaSuccess)
60          throw std::runtime_error(cudaGetErrorString(err));
61
62      return y;
63    }
64    """
65
66    cpp_decls = r"""
67    torch::Tensor relu_bias_add_cuda(torch::Tensor x, torch::Tensor bias);
68    """
69
70    # Compile / load the CUDA extension
71    relu_bias_add = load_inline(
72        name="relu_bias_add",
73        cpp_sources=cpp_decls,
74        cuda_sources=cuda_src,
75        functions=["relu_bias_add_cuda"],
76        verbose=False,
77    )
```

```
78
79   # ————————————————————————————————————
80   # Optimised model using the fused CUDA kernel
81   # ————————————————————————————————————
82   class ModelNew(nn.Module):
83       """"""
84       Same computation as original Model but with a fused
85       ReLU + bias−add CUDA kernel for higher performance.
86       """"""
87       def __init__(self, in_channels, out_channels, kernel_size, bias_shape):
88           super().__init__()
89           self.conv = nn.Conv2d(in_channels, out_channels, kernel_size)
90           # Bias after ReLU (per−channel, 11 spatial)
91           self.bias = nn.Parameter(torch.randn(bias_shape))
92           self._relu_bias_add = relu_bias_add
93
94       def forward(self, x):
95           x = self.conv(x)
96           # Fused kernel:  y = relu(x) + bias
97           x = self._relu_bias_add.relu_bias_add_cuda(x, self.bias)
98           return x
99
100
101  # ————————————————————————————————————
102  # Helper functions (unchanged)
103  # ————————————————————————————————————
104  batch_size = 32
105  in_channels  = 32
106  out_channels = 64
107  height = width = 64
108  kernel_size = 3
109  bias_shape = (out_channels, 1, 1)
110
111  def get_inputs():
112      return [torch.rand(batch_size, in_channels, height, width, device="cuda")]
113
114  def get_init_inputs():
115      return [in_channels, out_channels, kernel_size, bias_shape]
116  ```
```

Then the Judge got the runtime log and current kernel to analyze the error. However, the runtime log tracked back to another file, which we used to compare results. Without any information about this file, our workflow failed in the every turn of the 10-round iteration. This result reveals the challenge in automatically developing kernels in multi-file situation.

## B  DETAILS OF EVALUATION

**Collecting Test Cases.**  To obtain reliable and representative test cases, we do not directly use the single default input shape provided by KernelBench. Instead, for each task, we query the GPT-4o API to generate ten diverse input shapes, ranging from moderately sized tensors (e.g., 4096*4096) to shapes that utilize a substantial portion of the RTX 6000 GPU's memory capacity (e.g., 16384*16384), as shown in the Table 6. This ensures that both correctness and performance are evaluated across a broad spectrum of realistic workloads and prevents the evaluation from being overly influenced by small-shape cases.

**Correctness Evaluation.** We then evaluate correctness through a two-stage procedure consisting of compilation and execution. In the compilation stage, we verify that the generated kernel is syntactically valid and can be successfully compiled into executable CUDA code. In the execution stage, we

| Task | Size in KernelBench | Perf | Max Size in Test | Perf | Change in Size |
|------|---------------------|------|------------------|------|----------------|
| Level 1 Task 8 | M = 8205
N = 2949
K = 5921 | 0.996× | M = 32820
N = 11796
K = 23684 | 0.993× | 64× |
| Level 1 Task 15 | M = 4096
N = 4096 | 3.063× | M = 16384
N = 16384 | 3.385× | 16× |
| Level 2 Task 21 | batch_size = 128
in_channels = 8
out_channels = 32
height = width = 256
kernel_size = 3
num_groups = 8 | 1.531× | batch_size = 128
in_channels = 8
out_channels = 32
height = width = 512
kernel_size = 3
num_groups = 8 | 1.449× | 4× |
| Level 2 Task 75 | batch_size = 1024
in_features = 8192
out_features = 8192
num_groups = 512 | 1.030× | batch_size = 1024
in_features = 19384
out_features = 19384
num_groups = 1024 | 1.017× | 8× |
| Level 3 Task 18 | batch_size = 64
input_channels = 3
height = 512
width = 512
num_classes = 1000 | 2.008× | batch_size = 64
input_channels = 3
height = 1024
width = 1024
num_classes = 1000 | 2.040× | 4× |

Table 6: Performance under KernelBench input sizes and maximum test sizes.

run the kernel on all ten input shapes and compare its outputs with those produced by the PyTorch reference implementation under the same inputs. A kernel is considered correct only if it successfully compiles and its numerical outputs are within a tolerance of 0.0001 for all test cases, which is a commonly adopted criterion (Ouyang et al., 2025; Lange et al., 2025; Baronio et al., 2025).

**Performance Evaluation.** Finally, we assess optimization performance using only the largest input shape in the generated test cases. The reason to select the largest is to aligns with the goal of CUDA kernel optimization, which is primarily motivated by large-scale workloads such as those found in LLM inference and training. For each task, we profile the candidate kernel on the largest shape. Then the Judge agent makes refinement decisions based on hardware feedback. After N refinement rounds, we select the most efficient correct kernel as the final result. When reporting speedup over the PyTorch baseline, we also use the largest input shape to ensure that GPU computation dominates runtime, instead of the PyTorch framework or OS overhead.

## C   ANALYSIS OF TIME COST IN CUDAFORGE

In this section, we provide an analysis of time cost in CudaForge. As shown in Table 7, for a typical KernelBench task with ten refinement rounds, the 25-minute end-to-end runtime is dominated by Nsight Compute profiling, which takes approximately 10–12 minutes in total. Kernel compilation accounts for 2–3 minutes, while LLM inference contributes about 9–11 minutes across all rounds. This breakdown shows that the overall runtime is determined primarily by profiling rather than model latency or compilation overhead. Since kernels are independent and can be processed concurrently, CudaForge scales effectively to larger codebases, with throughput largely governed by the degree of parallelism available for profiling rather than limitations of the framework itself.

| Category | Time |
|---|---|
| Nsight Compute profiling | 10–12 minutes |
| LLM inference | 9–11 minutes |
| Kernel compilation | 2–3 minutes |

Table 7: Runtime breakdown of major components.

# D  PROMPT

## D.1  SEED PROMPT FOR CODER(ONE-SHOT BASELINE PROMPT FROM KERNELBENCH)

We adopt the *One-shot Baseline Prompt* introduced in KERNELBENCH as our initial seed prompt for first round generation of all the baselines and our method. The full prompt is shown below.

```
1   You write custom CUDA kernels to replace the pytorch operators in the given architecture to
2   get speedups.You have complete freedom to choose the set of operators you want to replace.
3   You may make the decision to replace some operators with custom CUDA kernels and leave
4   others unchanged. You may replace multiple operators with custom implementations,
5   consider operator fusion opportunities (combining multiple operators into a single kernel, for
6   example, combining matmul+relu), or algorithmic changes (such as online softmax). You are
7   only limited by your imagination.
8
9   Here an example to show you the syntax of inline embedding custom CUDA operators in
    torch:
10  The example given architecture is:
11  ```
12  {few_base}
13  ```
14  The example new arch with custom CUDA kernels looks like this:
15  ```
16  {few_new}
17  ```
18
19  You are given the following architecture:
20
21  ```python
22  {arch_src}
23  ```
24  Optimize the architecture named Model with custom CUDA operators! Name your optimized
25  output architecture ModelNew. Output the new code in codeblocks. Please generate real
26  code, NOT pseudocode, make sure the code compiles and is fully functional. Just output
27  the new model code, no other text, and NO testing code!
```

## D.2  PROMPT FOR JUDGE

In our prompt design for the Judge agent, we place the role specification and output schema in the system prompt. The input prompt only supplies per-round context(runtime information, NCU metrics, error_log). The system prompt is fixed; only the input prompt content changes each round.

The system prompt for cuda kernel optimization:

```
1   You are a senior CUDA performance engineer. Read the target GPU spec, the PyTorch
2   reference code, the current CUDA candidate, and the Nsight Compute metrics. Then identify
    **exactly one** highest−impact speed bottleneck by 3−4 most important metrics, propose **
    exactly one** optimization method and propose a modification plan. Be surgical and metrics−
    driven.
```

```
3
4   Rules:
5   − Return **one and only one** optimization method  the largest expected speedup.
6   − Prefer changes that directly address measured bottlenecks (occupancy limits,
7     memory coalescing, smem bank conflicts, register pressure, long/short scoreboard
8     stalls, tensor−core underutilisation, etc.).
9   − Keep fields brief; avoid lists of alternatives, disclaimers, or generic advice.
10
11  Output format (JSON):
12  ```json
13  {
14    "bottleneck": "<max 30 words>",
15    "optimization method": "<max 35 words>",
16    "modification plan": "<max 35 words>"
17  }
18  """"
```

The input prompt for optimization:

```
1   # Target GPU
2   GPU Name: {gpu_name}
3   Architecture: {gpu_arch}
4   Details:
5   {gpu_items}
6
7
8   # Pytorch Reference
9   {python_code}
10
11
12  # CUDA candidate
13  ```python
14  {CUDA_CODE}
15  ```
16
17  # Nsight Compute metrics (verbatim)
18  {NCU_METRICS}
19
20  Read everything and follow the Rules exactly. Return the JSON in the specified format.
```

The system prompt for kernel correction:

```
1   You are a senior CUDA + PyTorch correctness auditor. Your job is to read a PyTorch
    reference and a CUDA candidate and report exactly one most critical correctness issue in the
    CUDA code that would cause a behavioral mismatch vs. the PyTorch reference. Be terse and
    precise.
2
3   Rules:
4
5   Return one and only one issue  the single highest−impact problem.
6
7   Prefer semantic/correctness issues over micro−optimizations or style.
8
9   If multiple issues exist, pick the one that most changes outputs or gradients.
10
11  If nothing clearly wrong is found, say it explicitly.
12
13  Keep each field brief; avoid extra commentary, lists, or alternatives.
```

```
14
15   Output format (JSON):
16   ```json
17   {
18      "critical_issue": "<max 20 words>",
19      "why_it_matters": "<max 35 words>",
20      "minimal_fix_hint": "<max 20 words>"
21   }
22   ```
```

The input prompt for kernel repair:

```
1    You are given:
2
3    ERROR_LOG:
4    {ERROR_LOG}
5
6    PyTorch reference (ground truth):
7
8    {PYTORCH_CODE}
9
10   CUDA candidate (to audit):
11
12   {CUDA_CODE}
13
14
15   Follow the Rules and produce the JSON exactly in the specified format.
```

## D.3 PROMPT FOR CODER

For the Coder, we use the default system prompt and put all task details in the input prompt. This keeps the agent simple and fully context-driven. .

The prompt for kernel optimization:

```
1    # Target GPU
2    GPU Name: {gpu_name}
3    Architecture: {gpu_arch}
4    Details:
5    {gpu_items}
6
7    You are a CUDA−kernel optimization specialist.
8
9    Analyze the provided architecture and **strictly apply the following STRATEGY** to
     produce an improved CUDA kernel.
10
11   ```python
12   {CUDA_CODE}
13   ```
14
15   [optimization instructions]
16   {optimization_suggestion}
17
18   GOAL
19
20   − Improve latency and throughput on the target GPU.
21   − Maintain correctness within atol=1e−4 or rtol=1e−4.
22   − Preserve the public Python API (same inputs/outputs, shapes, dtypes).
```

```
23
24
25   OUTPUT RULES (STRICT)
26   1. Inside the block, follow **exactly** this order:
27      1. Imports  'torch', 'torch.nn', 'load_inline'.
28      2. 'source'  triplequoted CUDA string(s) (kernel + host wrapper).
29      3. 'cpp_src'  prototypes for *all* kernels you expose.
30      4. **One** 'load_inline' call per kernel group.
31      5. 'class ModelNew(nn.Module)'  mirrors original inputs/outputs but calls
32         your CUDA kernels.
33   2. **Do NOT include** testing code, 'if __name__ == "__main__"', or extra prose.
34
35   ```python
36   # <your corrected code>
37   ```
```

The prompt for kernel correction:

```
1    You are a senior CUDA−extension developer.
2    Your job is to **FIX** the compilation or runtime errors in the Python script
3    shown below.
4
5    OUTPUT RULES (STRICT)
6    1. Inside the block, follow **exactly** this order:
7       1. Imports  'torch', 'torch.nn', 'load_inline'.
8       2. 'source'  triplequoted CUDA string(s) (kernel + host wrapper).
9       3. 'cpp_src'  prototypes for *all* kernels you expose.
10      4. **One** 'load_inline' call per kernel group.
11      5. 'class ModelNew(nn.Module)'  mirrors original inputs/outputs but calls
12         your CUDA kernels.
13   2. **Do NOT include** testing code, 'if __name__ == "__main__"', or extra prose.
14
15
16   ERROR LOG
17
18   {ERROR_LOG}
19
20
21   OLD CODE (read−only)
22
23   {CUDA_CODE}
24
25
26   Main Critical Problem
27
28   {Problem}
29
30   ```python
31   # <your corrected code>
32   ```
```

# E DETAIL FOR THE NCU METRICS

## E.1 WHY CHOOSE NCU SUBSET METRICS?

We find that exposing large models to the full NCU metric set overwhelms them, reducing the accuracy and stability of their optimization suggestions and degrading Judge output quality. We illustrate this with following specific case study.

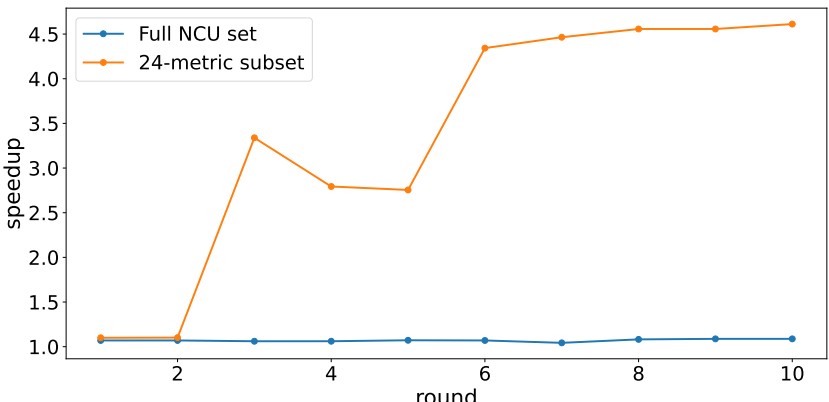

Figure 5: Generated kernel performance comparison between Full-metrics input(blue line) vs. Metrics subset input(orange line) for Judge agent on Level 2 task 51.

As shown in Figure 5, we compare two configurations on a representative task. (i) The Judge agent receives all raw NCU metrics in each round and provides feedback to the Coder agent. (ii) The Judge agent receives only the 24 key metrics selected by our method and captures only 3-4 critical metrics in each round to generate optimization suggestion.

Within 10 iterative rounds, the Judge agent using full NCU metrics fails to guide the Coder agent toward better kernels. The performance of the generated kernels remains almost unchanged, despite repeated interactions. In contrast, when the Judge agent is restricted to use subset metrics, the performance of generated kernels varies across rounds and is often improved. This process eventually leads to a substantial speedup.

In addition, we carefully reviewed how the Judge identified bottlenecks and proposed optimizations in both configurations, full set vs. 24-subset. We select one Judge's response from the optimization round to better illustrate the difference in judge output, with full NCU and with a subset of NCU as judge input.

Response of using the full NCU metrics set:

```
1  {"bottleneck": "Long−scoreboard stalls from re−loading GEMM output out of DRAM (
      gemm_out round−trip dominates cycles).",
2    "optimization method": "Fuse the Linear GEMM and the current epilogue into one FP16
      tensor−core CUTLASS kernel that performs subtract, row−mean, GELU and residual add in−
      register, avoiding global−memory re−reads.",
3    "modification plan": "Write CUTLASS custom epilogue functor; use cublasLt or cutlass::
      gemm_universal to compute GEMM+A, keep accumulators in registers, warp−reduce to mean
      , apply GELU, add x_orig, directly store final output; expose via new PyTorch extension."}
```

Response of using the 24 key NCU subset:

```
1  {"bottleneck": "Kernel is DRAM−bound (102.9 % peak); 57 % long−scoreboard stalls from
      second pass reading original_x.",
```

```
2    "optimization method": "Fuse the second loop: cache each threads original_x elements in
     registers during diff accumulation, then add gelu_val and store, eliminating the extra BF
     global read.",
3    "modification plan": "Replace two for−loops with one: load linear_out, subtract, and
     original_x together, keep original_x[16] in per−thread array, finish reduction, add gelu_val,
     write to output; ~−4 MB traffic per batch, >30 % speedup."}
```

Based on these two responses, we find that judge with full set NCU mertics tends to misidentify the true bottleneck. The judge with full set NCU metrics attributes the bottleneck to re-loading gemm_out and recommends a monolithic CUTLASS epilogue that performs row-mean/GELU/residual in registers. This diagnosis is misaligned with our kernel's access pattern and is hard to realize for general shapes due to cross-tile aggregation. In contrast, the judge with 24-key subset correctly identifies a DRAM-bound kernel dominated by the second pass over x_orig, and proposes a one-pass rewrite that caches x_orig in registers during the first traversal and writes back after GELU, eliminating an entire B×F global memory read. This change is lightweight, architecture-agnostic, and yields consistent speedups (e.g., about 4 MB less traffic per batch, more than 30% in our setting).

### E.2 TOP-20 NCU METRICS EXAMPLE

This section reports, for several example tasks, the Top-20 Nsight Compute (NCU) metrics most correlated with runtime, ranked by the absolute value of the Pearson correlation coefficient. Here, runtime refers to the kernel's execution time. When the correlation coefficient is positive, larger metric values typically imply longer execution time; when it is negative, larger metric values typically imply shorter execution time. All metric names follow their original name in NCU.

Table 8: Task-Conv2D: Pearson correlation with runtime (Top-20).

| Metric Name | Correlation | Abs Correlation |
|---|---|---|
| sm__cycles_active.avg | 1.000 000 | 1.000 000 |
| gpc__cycles_elapsed.max | 1.000 000 | 1.000 000 |
| launch__occupancy_limit_shared_mem | 0.945 507 | 0.945 507 |
| dram_bytes.sum.per_second | −0.924 251 | 0.924 251 |
| gpu__dram_throughput.avg.pct_of_peak_sustained_elapsed | −0.924 155 | 0.924 155 |
| smsp__inst_executed.avg | 0.916 287 | 0.916 287 |
| smsp__inst_executed.sum | 0.916 287 | 0.916 287 |
| smsp__inst_issued.avg | 0.916 262 | 0.916 262 |
| smsp__inst_issued.sum | 0.916 262 | 0.916 262 |
| lts__t_sector_hit_rate.pct | 0.839 237 | 0.839 237 |
| smsp__sass_average_branch_targets_threads_uniform.pct | 0.810 334 | 0.810 334 |
| lts__throughput.avg.pct_of_peak_sustained_elapsed | −0.787 261 | 0.787 261 |
| smsp__inst_executed_op_branch.sum | 0.746 483 | 0.746 483 |
| launch__grid_size | 0.745 917 | 0.745 917 |
| lltex__t_sector_hit_rate.pct | 0.728 356 | 0.728 356 |
| gpc__cycles_elapsed.avg.per_second | 0.728 053 | 0.728 053 |
| dram__cycles_elapsed.avg.per_second | 0.665 784 | 0.665 784 |
| launch__waves_per_multiprocessor | 0.627 478 | 0.627 478 |
| launch__thread_count | 0.627 478 | 0.627 478 |
| launch__shared_mem_per_block_static | −0.610 501 | 0.610 501 |

Table 9: Task-SpMM: Pearson correlation with runtime (Top-20).

| Metric Name | Correlation | Abs Correlation |
|---|---|---|
| gpc__cycles_elapsed.max | 0.999 993 | 0.999 993 |
| sm__cycles_active.avg | 0.998 432 | 0.998 432 |
| gpu__compute_memory_request_throughput.avg.pct_... | −0.967 284 | 0.967 284 |
| gpu__compute_memory_throughput.avg.pct_of_peak_... | −0.964 455 | 0.964 455 |
| lts__t_sector_hit_rate.pct | 0.951 201 | 0.951 201 |
| dram_bytes.sum.per_second | −0.926 134 | 0.926 134 |
| gpu__dram_throughput.avg.pct_of_peak_sustained_... | −0.925 856 | 0.925 856 |
| lltex__throughput.avg.pct_of_peak_sustained_active | 0.871 262 | 0.871 262 |
| sm__inst_executed.avg.per_cycle_elapsed | −0.837 675 | 0.837 675 |
| smsp__issue_inst0.avg.pct_of_peak_sustained_active | 0.837 284 | 0.837 284 |
| smsp__issue_active.avg.pct_of_peak_sustained_... | −0.837 284 | 0.837 284 |

| Metric Name | Correlation | Abs Correlation |
|---|---|---|
| smsp__issue_active.avg.per_cycle_active | −0.837 283 | 0.837 283 |
| sm__inst_issued.avg.per_cycle_active | −0.836 185 | 0.836 185 |
| sm__inst_issued.avg.pct_of_peak_sustained_active | −0.836 185 | 0.836 185 |
| sm__inst_executed.avg.per_cycle_active | −0.836 160 | 0.836 160 |
| sm__instruction_throughput.avg.pct_of_peak_sust... | −0.806 478 | 0.806 478 |
| smsp__average_warp_latency_per_inst_issued.ratio | 0.802 793 | 0.802 793 |
| smsp__average_warps_active_per_inst_executed.ratio | 0.802 777 | 0.802 777 |
| derived__smsp__inst_executed_op_branch_pct | −0.728 768 | 0.728 768 |
| smsp__warps_eligible.avg.per_cycle_active | −0.630 772 | 0.630 772 |

### E.3 KEY SUBSET OF 24 NCU METRICS

The table below lists the exact 24 metrics in our task-agnostic key subset.

Table 10: The 24-metric key subset.

| # | Metric Name |
|---|---|
| 1 | sm__cycles_active.avg |
| 2 | sm__warps_active.avg.pct_of_peak_sustained_active |
| 3 | launch__occupancy_limit_blocks |
| 4 | launch__occupancy_limit_registers |
| 5 | launch__occupancy_limit_shared_mem |
| 6 | launch__registers_per_thread |
| 7 | sm__inst_executed.sum |
| 8 | sm__inst_executed_pipe_fp32.avg.pct_of_peak_sustained_active |
| 9 | sm__inst_executed_pipe_tensor.avg.pct_of_peak_sustained_active |
| 10 | dram__bytes_read.sum |
| 11 | dram__bytes_write.sum |
| 12 | dram__throughput.avg.pct_of_peak_sustained_elapsed |
| 13 | dram__bytes.sum.per_second |
| 14 | gpu__dram_throughput.avg.pct_of_peak_sustained_elapsed |
| 15 | l1tex__t_sector_hit_rate.pct |
| 16 | l1tex__throughput.avg.pct_of_peak_sustained_active |
| 17 | lts__t_sector_hit_rate.pct |
| 18 | lts__throughput.avg.pct_of_peak_sustained_active |
| 19 | smsp__warp_issue_stalled_memory_dependency_per_warp_active.pct |
| 20 | smsp__warp_issue_stalled_short_scoreboard_per_warp_active.pct |
| 21 | smsp__warp_issue_stalled_long_scoreboard_per_warp_active.pct |
| 22 | smsp__warp_issue_stalled_barrier_per_warp_active.pct |
| 23 | smsp__warp_issue_stalled_branch_resolving_per_warp_active.pct |
| 24 | smsp__sass_average_branch_targets_threads_uniform.pct |

## F CUDA-L1

In our replication efforts, we found that the authors of CUDA-L1 released only the final, generated kernels for each task. After carefully studying these cases, we identified several interesting findings.

First, We found that CUDA-L1 tends to emphasize PyTorch-level optimizations rather than generating and refining custom CUDA kernels. This pattern also emerged as the most frequent issue in their provided case. Although CUDA-L1 reports the top-10 cases with the largest speedups, our review shows that nine of these ten final solutions do not use custom CUDA kernels; instead, they rely heavily on official PyTorch implementations.

This is the top-ranked entry in their *KernelBench Tasks Ranked by RL-CUDA1 Acceleration (Top-10)*: Level-2 Task 83, with a reported $120.3\times$ speedup

```
1  import torch
2  import torch.nn as nn
3
4  class ModelNew(nn.Module):
5      """
6      Optimized implementation of a model that performs a 3D convolution,
7      applies Group Normalization, minimum, clamp, and dropout.
8
9      Args:
```

```
10        in_channels (int): Number of input channels
11        out_channels (int): Number of output channels
12        kernel_size (int or tuple): Size of the convolving kernel
13        groups (int): Number of groups for GroupNorm
14        min_value (float): Minimum value for clamp operation
15        max_value (float): Maximum value for clamp operation
16        dropout_p (float): Dropout probability
17        """
18    def __init__(self, in_channels, out_channels, kernel_size, groups, min_value, max_value,
          dropout_p):
19        super(ModelNew, self).__init__()
20        # Store the original layers for parameter compatibility
21        self.conv = nn.Conv3d(in_channels, out_channels, kernel_size)
22        self.norm = nn.GroupNorm(groups, out_channels)
23        self.dropout = nn.Dropout(dropout_p)
24        self.min_value = min_value
25        self.max_value = max_value
26        self.dropout_p = dropout_p
27
28        # Pre-compute convolution parameters for output shape calculation
29        if isinstance(kernel_size, int):
30            self.kernel_size = (kernel_size, kernel_size, kernel_size)
31        else:
32            self.kernel_size = kernel_size
33
34        self.stride = self.conv.stride
35        self.padding = self.conv.padding
36        self.dilation = self.conv.dilation
37
38    def forward(self, x):
39        x = self.conv(x)
40        x = self.norm(x)
41        x = torch.minimum(x, torch.tensor(self.min_value, device=x.device))
42        x = torch.clamp(x, min=self.min_value, max=self.max_value)
43        x = self.dropout(x)
44        return x
45
46  # Keep ALL hyperparameters EXACTLY as shown in the reference implementation
47  batch_size = 128
48  in_channels = 3
49  out_channels = 16
50  depth, height, width = 16, 32, 32
51  kernel_size = 3
52  groups = 8
53  min_value = 0.0
54  max_value = 1.0
55  dropout_p = 0.2
56
57  def get_inputs():
58      return [torch.randn(batch_size, in_channels, depth, height, width)]
59
60  def get_init_inputs():
61      return [in_channels, out_channels, kernel_size, groups, min_value, max_value, dropout_p]
```

The second-ranked case is Level-1 Task 12 (Matmul with diagonal matrices), with a reported 64.4×
speedup

```
1   # diag_mm_compare.py
```

```python
2   import time
3   import math
4   import torch
5   import torch.nn as nn
6   import torch.nn.functional as F
7
8   # ————————————————————————————
9   # Reference implementation
10  # ————————————————————————————
11  class Model(nn.Module):
12      """
13      Simple model that performs a matrix multiplication of a diagonal matrix with another
        matrix.
14      C = diag(A) * B
15      """
16      def __init__(self):
17          super(Model, self).__init__()
18
19      def forward(self, A, B):
20          """
21          Args:
22              A (torch.Tensor): 1D tensor, diagonal entries. Shape: (N,)
23              B (torch.Tensor): 2D tensor. Shape: (N, M)
24          Returns:
25              torch.Tensor: (N, M)
26          """
27          return torch.diag(A) @ B
28
29
30  # ————————————————————————————
31  # Optimized implementation
32  # ————————————————————————————
33  class ModelNew(nn.Module):
34      """
35      Optimized model that performs a matrix multiplication of a diagonal matrix with another
        matrix.
36      C = diag(A) * B
37      """
38      def __init__(self):
39          super(ModelNew, self).__init__()
40
41      def forward(self, A, B):
42          """
43          Args:
44              A (torch.Tensor): 1D tensor, diagonal entries. Shape: (N,)
45              B (torch.Tensor): 2D tensor. Shape: (N, M)
46          Returns:
47              torch.Tensor: (N, M)
48          """
49          # Equivalent to torch.diag(A) @ B, but avoids forming the full diagonal matrix
50          return B * A.unsqueeze(1)
51
52
53  # ————————————————————————————
54  # Hyperparameters & inputs
55  # ————————————————————————————
56  M = 4096
57  N = 4096
58
```

```
59  def get_inputs(device=None, dtype=torch.float32):
60      A = torch.randn(N, device=device, dtype=dtype)
61      B = torch.randn(N, M, device=device, dtype=dtype)
62      return [A, B]
63
64  def get_init_inputs():
65      return []  # No special initialization inputs needed
```

In addition, we observed many reported speedups that are effectively equal to one (clustered around 1.00, typically within ±5%). A closer inspection shows that, in these cases, the system falls back to the original PyTorch operator when the custom kernel fails to compile, which naturally yields no measurable speedup.

For example, below is the forward method from the final solution for KernelBench Level-1 Task 3 generated by CUDA-L1. This code get from the CUDA-L1's official Github. We observe that the method first attempts to call a *custom CUDA kernel*; however, upon any compilation failure or exception, it immediately falls back to `torch.bmm(A, B)`. Crucially, `torch.bmm(A, B)` is exactly the operator that this task asks to be replaced by a custom kernel, meaning the fallback undermines the task's objective. This explains why the reported speedup is only 1.006×.

```python
1   def forward(self, A: torch.Tensor, B: torch.Tensor) -> torch.Tensor:
2       """
3       Performs batched matrix multiplication.
4
5       Args:
6           A: Input tensor of shape (batch_size, m, k).
7           B: Input tensor of shape (batch_size, k, n).
8
9       Returns:
10          C: Output tensor of shape (batch_size, m, n).
11      """
12      # Fall back to torch.bmm if CUDA module failed to load
13      if ModelNew._cuda_module is None:
14          return torch.bmm(A, B)
15
16      # Check if inputs are on CUDA
17      if not A.is_cuda or not B.is_cuda:
18          A = A.cuda() if not A.is_cuda else A
19          B = B.cuda() if not B.is_cuda else B
20
21      # Ensure inputs are contiguous and float32
22      A = A.contiguous().float()
23      B = B.contiguous().float()
24
25      # Use custom CUDA kernel
26      try:
27          result = ModelNew._cuda_module.batched_matmul(A, B)
28          if not A.is_cuda:
29              result = result.cpu()
30          return result
31      except Exception as e:
32          print(f"Error in custom kernel: {e}, falling back to torch.bmm")
33          return torch.bmm(A, B)
```

## G    DETAILS OF BENCHMARK

### G.1    KERNELBENCH

**KernelBench** is a standardized benchmark designed to evaluate the capability of large language models (LLMs) in CUDA kernel generation and optimization. It consists of 270 tasks across four levels of increasing difficulty, of which Levels 1–3 (250 tasks in total) are commonly adopted for evaluation. Each task provides a PyTorch reference implementation $f_{T_i}$ together with fixed input–output specifications, enabling automated correctness and performance validation.

- **Level 1 (Basic Operators):** Contains simple, low-level operators such as matrix multiplication, element-wise operations, and reductions. These tasks primarily test the ability to generate functionally correct CUDA kernels.
- **Level 2 (Composite Operations):** Involves multi-step operator combinations, requiring the model to compose multiple CUDA primitives and manage intermediate memory efficiently. These tasks test the capacity for more complex code synthesis.
- **Level 3 (End-to-End Models):** Includes challenging kernels derived from full neural network architectures such as AlexNet, VGG, and ResNet components. These tasks assess the ability to produce efficient, large-scale kernels under realistic deep learning workloads.
- **Level 4 (Optional):** The full benchmark also defines an advanced level with additional research-oriented tasks, but this is less frequently adopted due to its complexity and lack of standardized evaluation setups.

KernelBench has become a widely used benchmark in recent work on LLM-based code generation (Team, 2025; Baronio et al., 2025; Lange et al., 2025), as it provides a controlled and reproducible environment to measure both *correctness* (functional equivalence to PyTorch) and *efficiency* (execution speed relative to PyTorch). In our study, we adopt all Level 1–3 tasks, following prior work, to ensure fair comparison across baselines.

### G.2    OUR STRATIFIED RANDOM SUBSET $\mathcal{D}^*$

While our main evaluation is conducted on the full KernelBench Level 1–3 benchmark (250 tasks in total), we additionally construct a stratified subset $\mathcal{D}^*$ to enable detailed analysis and fair comparison with prior work such as Kevin.

The construction of $\mathcal{D}^*$ follows two principles: (1) **Coverage across difficulty levels.** Since KernelBench is stratified by increasing task complexity (Level 1: single-operator tasks, Level 2: multi-step fused operators, Level 3: full network components), we ensure that the sampled subset preserves the relative distribution of difficulty. (2) **Diversity of task types.** Within each level, we sample tasks uniformly across different operator categories (e.g., elementwise ops, reductions, convolutions, fused blocks) so that the subset remains representative of the overall benchmark.

Concretely, we perform stratified random sampling with a fixed 10% ratio for each level, resulting in a subset of 10 tasks from Level 1, 10 tasks from Level 2, and 5 tasks from Level 3, for a total of 25 tasks. For reproducibility, the exact task IDs included in $\mathcal{D}^*$ are:

- **Level 1 (10 tasks):** 13, 10, 16, 29, 35, 72, 7, 89, 93, 34
- **Level 2 (10 tasks):** 17, 19, 40, 3, 13, 21, 38, 28, 26, 34
- **Level 3 (5 tasks):** 5, 18, 32, 41, 21

## USAGE OF LLM

During the preparation of this paper, we employed large language models (LLMs) solely for **textual assistance**, including grammar correction, stylistic refinement, and clarity improvements. All core research contributions—including the design of `CudaForge`, implementation of experiments, and analysis of results—were conducted entirely by the authors. The LLM was not used to generate research ideas, experimental results, or any substantive content of the paper.

