# OpenReview forum: "CudaForge: An Agent Framework with Hardware Feedback for CUDA Kernel Optimization"
_ICLR.cc/2026/Conference — Submitted to ICLR 2026_

### Official Review · Reviewer_PHnN · 2025-10-14

**Soundness:** 3
**Presentation:** 3
**Contribution:** 2
**Rating:** 2
**Confidence:** 5

**Summary:**

This paper presents CudaForge, a training-free multi-agent framework for CUDA kernel optimization. By integrating GPU kernel performance metrics directly into its feedback loop, CudaForge efficiently refines generated kernels, achieving notable speedups while remaining cost-effective.

**Strengths:**

- The paper clearly identifies the limitations of prior work and demonstrates significant performance improvements over existing approaches.
- The evaluation is thorough and well-executed. In particular, the ablation study is clearly presented and highlights the key NCU metrics that drive performance gains in the generated CUDA kernels.
- The work clearly shows that incorporating NCU metrics is fundamental to improving CUDA kernel performance, and this insight has broader applicability to other agent-based optimization workflows.
- The inclusion of complete prompts for both the generator and judge models, along with detailed model specifications, makes it easy to produce the results.

**Weaknesses:**

- One of the paper's central claims is the use of a two-agent workflow, where one agent judges and the other codes. While the results do indicate that separating these roles yields better outcomes than a single agent handling both tasks, this observation is not particularly novel. Similar findings have been reported in prior work (e.g., [1]) and in domains beyond CUDA kernel generation. As a result, the paper’s unique contribution in this regard is unclear.
- Another claimed contribution is the exploration of which NCU metrics to incorporate into the judge's feedback. However, the identification and use of representative NCU metrics has been studied extensively in prior literature (e.g., [2], [3], [4]). This work does not appear to provide significant innovation beyond existing methods. A stronger positioning would involve explicitly situating the approach within this prior body of work, clarifying what is new, and citing relevant references to give readers a more complete perspective.
- The main takeaway seems to be that incorporating a subset of NCU metrics into feedback can improve kernel efficiency. While this is a useful observation, the paper offers limited additional insights or deeper analysis beyond this point.
- The paper suggests that reducing the number of NCU metrics prevents the coder from being overwhelmed. However, the evidence is conducted in a single task (Figure 5). It remains unclear whether this effect generalizes across a broader set of tasks.
    - The paper does not explore what the "sweet spot" is for t
he number of NCU metrics. Is trial-and-error the intended approach, or are there principled guidelines?
    - It is also unclear whether the observed performance gains stem from the quality of the selected metrics (i.e., stronger correlation with performance) or simply from reducing the volume of NCU metrics.
- Minor issues
    - In the section “Comparison with O3-10-O (optimization-only Judge)”, it would be helpful to include concrete numbers to quantify how much correctness is compromised.
    - On line 362, Appendix E is missing its reference.
    - There are inconsistencies in spelling: some plots use "optimisation" while the main text uses "optimization."

[1] Lange, Robert & Sun, Qi & Prasad, Aaditya & Faldor, Maxence & Tang, Yujin & Ha, David. (2025). Towards Robust Agentic CUDA Kernel Benchmarking, Verification, and Optimization. 10.48550/arXiv.2509.14279.
[2] S. Che et al., "Rodinia: A benchmark suite for heterogeneous computing," 2009 IEEE International Symposium on Workload Characterization (IISWC), Austin, TX, USA, 2009, pp. 44-54, doi: 10.1109/IISWC.2009.5306797. keywords: {Kernel;Multicore processing;Parallel processing;Application software;Yarn;Benchmark testing;Central Processing Unit;Energy consumption;Microprocessors;Computer architecture},
[3] B. Hu and C. J. Rossbach, "Altis: Modernizing GPGPU Benchmarks," 2020 IEEE International Symposium on Performance Analysis of Systems and Software (ISPASS), Boston, MA, USA, 2020, pp. 1-11, doi: 10.1109/ISPASS48437.2020.00011. keywords: {Runtime;Graphics processing units;Focusing;Production;Computer architecture;Benchmark testing;Hardware;GPGPU},
[4]  Che S, Skadron K. BenchFriend: Correlating the performance of GPU benchmarks: Correlating the performance of GPU benchmarks. The International Journal of High Performance Computing Applications. 2013;28(2):238-250. doi:10.1177/1094342013507960

**Questions:**

1. The paper adopts 24 NCU metrics, but the rationale for this specific number is unclear. Why is 24 considered optimal? Would a smaller subset (e.g., the top 10 most representative metrics) yield comparable performance? Conversely, is there evidence of a threshold beyond which adding more metrics begins to degrade kernel speedup—either gradually or abruptly? A sensitivity analysis would strengthen the argument here.
2. It remains ambiguous whether the observed kernel speedup improvements are primarily driven by the quality of the selected NCU metrics (i.e., their correlation with performance outcomes) or simply by the quantity of metrics included in the feedback. Clarifying this distinction would provide deeper insight into the mechanism behind the improvements.
3. Results in Table 4 suggest that the choice of Coder model has a stronger influence on the quality of generated code than the Judge. If this is the case, could a smaller or less capable Judge, when paired with a more powerful Coder, achieve comparable performance?

---

> ### Author Response · Authors · 2025-11-21
>
> Thanks for your time and effort in reviewing our work. We sincerely appreciate your recognition of our contributions in excellent performance, NCU incorporation, comprehensive evaluation, and efforts for reproductivity. Also we thank your valuable suggestions for our paper. Below, we address your concerns point by point and we’ll revise our paper according to your suggestions. We would appreciate it if you could let us know whether your concerns are addressed by our response.
>
> Question 1：
>
> One of the paper's central claims is the use of a two-agent workflow, where one agent judges and the other codes. While the results do indicate that separating these roles yields better outcomes than a single agent handling both tasks, this observation is not particularly novel. Similar findings have been reported in prior work (e.g., [1]) and in domains beyond CUDA kernel generation. As a result, the paper’s unique contribution in this regard is unclear.
>
> Overall, the primary contribution appears limited to simply embedding NCU metrics into the prompts, making the LLM produce faster kernels.
>
>
> [1] Lange, Robert & Sun, Qi & Prasad, Aaditya & Faldor, Maxence & Tang, Yujin & Ha, David. (2025). Towards Robust Agentic CUDA Kernel Benchmarking, Verification, and Optimization. 10.48550/arXiv.2509.14279.
>
> Answer 1：
>
> Thank you for your concerns. We would like to clarify the relationship between our CudaForge and other works, and demonstrate our unique contribution to this field.
>
> First, We would first like to clarify that [1] is concurrent work, explicitly acknowledged in lines 59–63 of our paper. According to the ICLR 2026 Reviewer Guidelines (https://iclr.cc/Conferences/2026/ReviewerGuide), authors are not required to compare against papers posted after July 24, 2025. Since [1] was released on arXiv on September 16, it should not be used as a basis for evaluating novelty or positioning. Moreover, the emergence of multi-agent frameworks in concurrent work underscores the relevance and necessity of our contribution to this newly developing research direction.
>
> Second, whether multi-agent frameworks are effective for LLM-based CUDA kernel generation and optimization was previously unexplored. While multi-agent systems have shown benefits in other domains, these findings cannot be assumed to transfer automatically. CUDA kernel generation is a domain with distinct constraints(e.g. hardware-level performance dynamics, memory hierarchy behaviors, and correctness requirements) which make the design and roles of the agents non-trivial. Therefore, demonstrating that a carefully constructed two-agent workflow (Coder + Judge) yields significant gains in this domain is a meaningful and novel contribution.
>
> Third, although our two-agent structure appears simple, this should not be considered as trivial. A multi-agent system does not yield improvements merely by “adding more agents.” Its effectiveness depends on (i) how roles are defined, (ii) how information flows between agents, and (iii) how external tools (in our case, Nsight Compute profiling) are integrated into the loop. CudaForge makes specific, deliberate design choices: the Coder focuses solely on generation while the Judge focuses solely on evaluation, separating the cognitive load; the Judge extracts and filters NCU metrics through a systematic methodology; and the agents interact in an iterative workflow explicitly aligned with human expert optimization. These design decisions directly contribute to CudaForge achieving state-of-the-art performance on KernelBench Levels 1–3, while maintaining low complexity, API usage, and runtime.
> Finally, we summarize the core contributions of our work:
>
>  (1) A training-free multi-agent framework for CUDA kernel generation and optimization that incorporates hardware execution feedback to drive targeted, rather than blind, refinement. This contribution is recognized by Reviewers wyaT, BMAj, and TYZF.
>
>  (2) A systematic methodology for integrating Nsight Compute profiling signals into the workflow, including selecting, filtering, and structuring metrics so they are actionable for the Judge agent. Reviewers wyaT, BMAj, and TYZF all highlight this as an important contribution.
>
>  (3) Comprehensive evaluation across KernelBench Levels 1–3 with extensive ablations demonstrating the contribution of each component. This is recognized by Reviewers PHnN, BMAj, and TYZF.
>
>  (4) Compared with other agentic approaches, CudaForge achieves significantly lower API cost and runtime while delivering superior performance, which is also recognized by Reviewer wyaT.

---

> ### Author Response · Authors · 2025-11-21
>
> Question 2：
>
> Another claimed contribution is the exploration of which NCU metrics to incorporate into the judge's feedback. However, the identification and use of representative NCU metrics has been studied extensively in prior literature (e.g., [2], [3], [4]). This work does not appear to provide significant innovation beyond existing methods. A stronger positioning would involve explicitly situating the approach within this prior body of work, clarifying what is new, and citing relevant references to give readers a more complete perspective.
>
> [2] A systematic methodology for integrating Nsight Compute profiling signals into the workflow, including selecting, filtering, and structuring metrics so they are actionable for the Judge agent. Reviewers wyaT, BMAj, and TYZF all highlight this as an important contribution.
>
> [3] Comprehensive evaluation across KernelBench Levels 1–3 with extensive ablations demonstrating the contribution of each component. This is recognized by Reviewers PHnN, BMAj, and TYZF.
>
> [4] Compared with other agentic approaches, CudaForge achieves significantly lower API cost and runtime while delivering superior performance, which is also recognized by Reviewer wyaT.
>
>
>
> Answer 2：
>
> Thank you for your comments. We would like to discuss the relationship between our work and [2][3][4], and claim our work is the first to systematically explore identification and use of representative NCU metrics in LLM Agent workflow.
>
> First, we must point out a **factual error** in the claim that “representative NCU metrics have been studied extensively in prior literature.” Nsight Compute (NCU) was released by NVIDIA in **September 2018**, as documented in NVIDIA’s official Nsight Compute Release History(https://developer.nvidia.com/nsight-compute-history). Two of the cited works, Rodinia (2009) [2] and BenchFriend (2013) [4], were **published five to nine years before NCU even existed**. It is therefore **impossible for these papers to have studied NCU metrics, profiler-guided CUDA optimization, or anything related to the Nsight Compute toolchain**. Regarding the third citation [3], we examined it carefully. This work focuses on modernizing a GPGPU benchmark suite by expanding workloads and improving compatibility with contemporary GPU architectures. It does not discuss Nsight Compute, does not analyze NCU metrics, and does not address CUDA kernel generation or performance refinement.
>
> In light of this, the assertion that our exploration of NCU metrics lacks novelty because it has been “studied extensively” is not supported and does not accurately reflect the existing literature. The cited works are unrelated to NCU metric selection and provide no evidence against the originality of our contribution. To the best of our knowledge, CudaForge is the first to systematically integrate NCU metric selection into an automated LLM-driven CUDA optimization loop. This novelty and contribution **has been recognized by all the other reviewers**.

---

> ### Author Response · Authors · 2025-11-21
>
> Question 3：
>
> The main takeaway seems to be that incorporating a subset of NCU metrics into feedback can improve kernel efficiency. While this is a useful observation, the paper offers limited additional insights or deeper analysis beyond this point.
>
> Answer 3：
>
> Thank you for your comments. We provide ablation studies and analysis in the following Answer 4, where we systematically analyze NCU metrics’ effect in CudaForge’s performance. Feel free to let us know if you have any questions!
>
>
> Question 4：
>
> The paper suggests that reducing the number of NCU metrics prevents the coder from being overwhelmed. However, the evidence is conducted in a single task (Figure 5). It remains unclear whether this effect generalizes across a broader set of tasks.
>
> The paper does not explore what the "sweet spot" is for the number of NCU metrics. Is trial-and-error the intended approach, or are there principled guidelines?
>
> It is also unclear whether the observed performance gains stem from the quality of the selected metrics (i.e., stronger correlation with performance) or simply from reducing the volume of NCU metrics.
>
> Answer 4：
>
> Thank you for your insightful comments. Constrained by page limit in ICLR submission, we don’t have space to show our full ablation study results so we put a case study in our paper. Actually we have observed that this is a general situation in our experiments. Here we can show more ablation study results for you. The ablation study is conducted on the subset D*, aligned with all the other ablation studies in the original paper. In these experiments, **we tried randomly selecting metrics, feeding full metrics, selecting less metrics than our design, and comparing with our 24 metrics.**
>
> **Table R1: Ablation study on NCU metric selection. Comparing full, random, and top-k subsets shows
> that concise and carefully chosen metrics (Top-24) provide strong overall performance, while Top-
> 20 offers slightly higher speedup with similar behavior.**
>
> | Methods                                       | Correctness | Performance | $Fast_1$ |
> |-----------------------------------------------|---------|-------------|-------|
> | CudaForge (our method – top 24 metrics)       | 100%    | 1.767       | 84%   |
> | CudaForge (full metrics)                      | 100%    | 1.414       | 80%   |
> | CudaForge (random 24 metrics)                 | 100%    | 1.655       | 76%   |
> | CudaForge (our method – top 10 metrics)       | 100%    | 1.644       | 80%   |
> | CudaForge (our method – top 5 metrics)        | 100%    | 1.641       | 76%   |
> | CudaForge (our method – top 20 metrics)       | 100%    | 1.827       | 88%   |
>
> According to the results, we can get 2 conclusions:
>
> First, directly feeding full metrics to judge will lead to low quality feedback and lower performance of the generated kernel.
>
> Second, selecting too few metrics or randomly selecting will constrain the search space of the judge, leading to less-effective feedback. And our design always achieves overall best performance in these ablation experiments, except performance of top 20 metrics, which is close to our top 24 metrics.
>
> We admit that whether 24 is just the optimal setting or “sweet spot”  can be further discussed in future work. But what we want to emphasize is that **our experiments proves we indeed need a metrics selection method for Agent** and we explored one possible solution with the correlation score. To this end, our correlation based rule is at least better than full feeding or randomly selecting, we expect better methods on this in future work. We have added it to the revised version.

---

> ### Author Response · Authors · 2025-11-21
>
> Question 5：
>
> Minor issues
> In the section “Comparison with O3-10-O (optimization-only Judge)”, it would be helpful to include concrete numbers to quantify how much correctness is compromised.
> On line 362, Appendix E is missing its reference.
> There are inconsistencies in spelling: some plots use "optimisation" while the main text uses "optimization."
>
> Answer 5：
>
> Thank you for your suggestions in our paper writing. For comparison with O3-10-O (optimization-only Judge), we have revised it in our paper. For Appendix E on line 362, we would like to clarify that this Appendix E means Appendix E in [1], so that we could not provide reference to Appendix E here. For “optimization”, we have changed all the optimisation to optimization.
>
> Question 6：
>
> The paper adopts 24 NCU metrics, but the rationale for this specific number is unclear. Why is 24 considered optimal? Would a smaller subset (e.g., the top 10 most representative metrics) yield comparable performance? Conversely, is there evidence of a threshold beyond which adding more metrics begins to degrade kernel speedup—either gradually or abruptly? A sensitivity analysis would strengthen the argument here.
>
> Answer 6：
>
> Thank you for your insightful comments. We provide ablation studies and deep analysis in Answer 4, including sensitivity analysis of NCU metrics.
>
> The result demonstrates that:
> **1. selecting a high quality subset of NCU metrics contributes to better performance.
> 2. the observed performance gains stem from quality of NCU metrics, not quantity.
> 3. the sensitivity analysis supports that too small will lead to suboptimal performance.**
> Our selection method could select a subset leading to optimal performance, and the method is robust: the performance wouldn’t sharply decline if we subtly change the number of metrics. We will provide results of selecting a larger NCU subset later.
>
>  Feel free to let us know if you have any questions!
>
> Question 7：
>
> It remains ambiguous whether the observed kernel speedup improvements are primarily driven by the quality of the selected NCU metrics (i.e., their correlation with performance outcomes) or simply by the quantity of metrics included in the feedback. Clarifying this distinction would provide deeper insight into the mechanism behind the improvements.
>
> Answer 7：
>
> Thank you for your insightful comments. We provide ablation studies and deep analysis in Answer 4, where we demonstrate whether quality or quantity of the selected NCU metrics leads to CudaForge's performance. **As a result, it is the quality of NCU metrics that is really important, aligning with our findings in the original paper.**
>
> Question 8：
>
> Results in Table 4 suggest that the choice of Coder model has a stronger influence on the quality of generated code than the Judge. If this is the case, could a smaller or less capable Judge, when paired with a more powerful Coder, achieve comparable performance?
>
> Answer 8：
>
> Thank you for the thoughtful question. Table 4 indeed suggests that the Coder model has a stronger influence on the final performance than the Judge. This observation is consistent with one of our key design goals: the Judge relies primarily on **rule-based NCU metric selection and structured hardware signals**, rather than on the intrinsic capability of the underlying language model. Because the Judge’s decisions are guided by explicit profiler-derived rules, its performance is less sensitive to model size.
>
> This also implies that the Judge can potentially be replaced with **a smaller and more cost-efficient model** (e.g., a 7B model) without significant loss in overall performance, offering a promising direction for reducing workflow cost. While we have not fully explored this substitution in the current paper due to computational constraints, we consider it an exciting avenue for future work and plan to evaluate lightweight Judges in follow-up experiments.

---

> ### Comment · Reviewer_PHnN · 2025-11-21
> **Further followup**
>
> I appreciate the authors for raising the concern regarding NCU metrics profiling, and I would like to take this opportunity to clarify the intention behind my question. The use of GPU metrics to categorize workloads has been studied extensively in prior work. Whether these metrics are collected through NCU, nvprof, GPGPU-sim, or other CUDA-related (and I apologize for any earlier imprecision in my terminology) tools is not the central issue. While I acknowledge that NCU provides access to metrics that may not be available in earlier tool suites, the broader concept of collecting low-level metrics to classify CUDA kernels predates the introduction of NCU. Relying solely on switching the collection tool to NCU may not, by itself, represent a novel contribution.
>
> Furthermore, employing Pearson correlation to categorize GPU workloads using low-level metrics is a well-established practice in the literature. My main point is to understand whether the proposed metrics collection process introduces any methodological innovation or substantive advancement beyond what has already been explored in existing studies.

---

> ### Author Response · Authors · 2025-11-22
>
> Thanks very much for your prompt response.
>
> First, our core contribution lies in enabling automated and customized GPU kernel generation and optimization through a multi-agent system, as summarized in Answer 1. To the best of our knowledge, we are the first to guide an agent to effectively utilize GPU profiling tools within this workflow. We demonstrate that, without such guidance, the agent cannot meaningfully interpret the information provided by profiling tools, leading to inferior performance. We explore a Pearson-correlation-based metric selection strategy for the judge agent and show that it yields clear improvements.
>
> Second, the use of Pearson correlation on low-level hardware metrics is not a well-established practice for guiding GPU kernel optimization. The reference, [4] Che S, Skadron K. BenchFriend: Correlating the performance of GPU benchmarks: Correlating the performance of GPU benchmarks. The International Journal of High Performance Computing Applications. 2013;28(2):238-250. , leverages only a small set of metrics to predict the performance of GPU applications, rather than to guide optimization decisions. Moreover, their approach relies on five high-level simulator metrics, whereas NVIDIA NCU exposes hundreds of low-level hardware metrics on real GPUs.  Identifying which metrics are relevant, and filtering them effectively to provide actionable feedback to an agent, is a central bottleneck in automated kernel development. The other cited work, Altis: Modernizing GPGPU Benchmark, is an application-level benchmark suite and does not discuss low-level hardware metrics at all.
>
> Finally, we would like to clarify and summarize: Providing a systematic method to integrate NCU metrics is really important for agentic workflows. As the first work to comprehensively explore this topic, the metric selection strategy, in our opinion, should be considered as novel. Moreover, the contributions of our work are not limited to the selection of NCU metrics. We have summarized them in Answer 1, and they are generally recognized by other reviewers.

---

> ### Author Response · Authors · 2025-11-26
>
> Dear Reviewer,
>
> I hope this message finds you well. As the discussion period is nearing its end with less than one week remaining, l want to ensure we have addressed all your concerns satisfactorily. If there are any additional points or feedbacks you'd like us to consider, please let us know. Your insights are invaluable to us, and we are eager to address any remaining issues to improve our work.
>
> Thank you for your time and effort in reviewing our paper!

---

### Official Review · Reviewer_BMAj · 2025-10-29

**Soundness:** 3
**Presentation:** 3
**Contribution:** 3
**Rating:** 6
**Confidence:** 4

**Summary:**

CudaForge proposes multi agent framework with iterative refinement to produce performant CUDA kernels. Authors propose use of two models working independently 1) coder: focuses on generating code given a prompt and optionally feedback, 2) judge: focuses on analysing execution and hardware feedback and generate cues for coder model to improve correctness or performance. Authors also provide a methodology to parse relevant information from hardware feedback and this helps in removing redundant and irrelevant metrics that might jeopardise producing good kernels. Authors have presented comparison with other baselines and good enough ablation study.

**Strengths:**

- Propose train-free iterative refinement based approach for CUDA kernel generation.
- Demonstrate the use of LLMs with different identities (coder & judge) in producing performant CUDA kernels.
- Provide a systematic methodology to extract and refine the output of Nsight profiler.
- Provide detailed analysis of related methods and bring forth interesting observations.
- Method has been shown to work across various frontier and open source models.

**Weaknesses:**

- Efficacy of this approach is not demonstrated by authors on popular but low resource languages such as Triton.
- Unlike other approaches, there is no clear methodology of evaluation specified in the paper. Precise evaluation setup is extremely important in such tasks.
- Performance measurement with native pytorch implementation without torch.compile does not reflect a comparison with a true baseline.

**Questions:**

- Does CudaForge scale to Triton programming? What performance improvements can be achieved there?
- How does CudaForge methodology evaluate speedup of generated kernels?
- How does this approach compare against implementations like AlphaEvolve/OpenEvolve?
- How does CudaForge performance metrics look like when compared against torch.compile version of pytorch?

---

> ### Author Response · Authors · 2025-11-21
>
> Thanks for your time and effort in reviewing our work. We sincerely appreciate your recognition of our contributions in effective multi-agent training-free framework, novel methodology on the NCU profiler, and comprehensive analysis and experiments. Also, we thank your valuable suggestions for our paper. Below, we address your concerns point by point, and we’ll revise our paper according to your suggestions. We would appreciate it if you could let us know whether your concerns are addressed by our response.
>
> Weakness 1:
>
> The efficacy of this approach is not demonstrated by the authors on popular but low-resource languages such as Triton.
>
> Answer 1：
>
> Thank you for raising this insightful point. We agree that Triton is an increasingly popular GPU programming language, and it is reasonable to ask whether our approach can generalize beyond CUDA.
>
> We would like to clarify that CudaForge itself is not tied to CUDA-specific settings. The two-agent workflow, where the Coder generates kernels and the Judge analyzes runtime and hardware feedback, is fundamentally language-agnostic. The Judge agent relies on profiling signals obtained after kernel execution; these signals can also be collected for Triton kernels (e.g., via Nsight Compute or other hardware-level profilers). Similarly, the Coder can be prompted to produce Triton code without modifying the overall algorithm. It also demonstrates the advantage in training free of CudaForge: We don’t need to train a new model to transfer to Triton optimization.
>
> We appreciate the importance of supporting Triton. We will add relevant citations on Triton to the revised paper. As part of ongoing work, we are extending CudaForge to Triton kernels. We will include Triton experiments in future extensions of this work, and if time permits, we will make our best effort to incorporate them into the revised version.

---

> ### Author Response · Authors · 2025-11-21
>
> Weakness 2:
>
> Unlike other approaches, there is no clear methodology of evaluation specified in the paper. Precise evaluation setup is extremely important in such tasks.
>
> Answer 2:
>
> Thank you for highlighting the importance of clearly specifying the evaluation methodology. We agree that precise evaluation procedures are essential in this line of work, and we provide a more structured description of our evaluation pipeline here, including collecting test cases, correctness evaluation, and performance evaluation.
>
> [Collecting test cases] To obtain reliable and representative test cases, we do not directly use the single default input shape provided by KernelBench. Instead, for each task, we query the GPT-4o API to generate ten diverse input shapes, ranging from moderately sized tensors (e.g., 4096x4096) to shapes that utilize a substantial portion of the RTX 6000 GPU’s memory capacity (e.g., 16384x16384), as shown in the Table R1. This ensures that both correctness and performance are evaluated across a broad spectrum of realistic workloads and prevents the evaluation from being overly influenced by small-shape cases.
>
> [Correctness evaluation] We evaluate correctness through a two-stage procedure consisting of compilation and execution. In the compilation stage, we verify that the generated kernel is syntactically valid and can be successfully compiled into executable CUDA code. In the execution stage, we run the kernel on all ten input shapes and compare its outputs with those produced by the PyTorch reference implementation under the same inputs. A kernel is considered correct only if it successfully compiles and its numerical outputs are within a tolerance of 1e−4 for all test cases, following standard practice in prior work [1, 2].
>
> [Performance Evaluation] Finally, we assess optimization performance using only the largest input shape in the generated test cases. The reason to select the largest is to align with the goal of CUDA kernel optimization, which is primarily motivated by large-scale workloads such as those found in LLM inference and training. For each task, we profile the candidate kernel on the largest shape. Then the Judge agent makes refinement decisions based on hardware feedback. After N refinement rounds, we select the most efficient correct kernel as the final result. When reporting speedup over the PyTorch baseline, we also use the largest input shape to ensure that GPU computation dominates runtime, instead of the PyTorch framework or OS overhead.
> In summary, our methodology guaranteed more robust evaluation, compared with just using one test case from KernelBench for evaluation.
>
> [1] Anne Ouyang, Simon Guo, Simran Arora, Alex L. Zhang, William Hu, Christopher R´e, and Azalia
> Mirhoseini. Kernelbench: Can llms write efficient gpu kernels?, 2025. URL https://arxiv.org/abs/2502.10517.
>
> [2] Robert Tjarko Lange, Qi Sun, Aaditya Prasad, Maxence Faldor, Yujin Tang, and David Ha. Towards
> robust agentic cuda kernel benchmarking, verification, and optimization, 2025. URL https://arxiv.org/abs/2509.14279.
>
> **Table R1. Performance under KernelBench input sizes and maximum test sizes**
>
> | Task            | Size in KernelBench                                      | Perf     | Max Size in Test                                      | Perf     | Change in Size |
> |-----------------|-----------------------------------------------------------|----------|--------------------------------------------------------|----------|----------------|
> | Level 1 Task 8  | M = 8205, N = 2949, K = 5921 | 0.996× | M = 32820, N = 11796, K = 23684 | 0.993× | 64× |
> | Level 1 Task 15 | M = 4096, N = 4096 | 3.063× | M = 16384, N = 16384 | 3.385× | 16× |
> | Level 2 Task 21 | batch_size = 128, in_channels = 8, out_channels = 32, height = width = 256, kernel_size = 3, num_groups = 8 | 1.531× | batch_size = 128, in_channels = 8, out_channels = 32, height = width = 512, kernel_size = 3, num_groups = 8 | 1.449× | 4× |
> | Level 2 Task 75 | batch_size = 1024, in_features = 8192, out_features = 8192, num_groups = 512 | 1.030× | batch_size = 1024, in_features = 19384, out_features = 19384, num_groups = 1024 | 1.017× | 8× |
> | Level 3 Task 18 | batch_size = 64, input_channels = 3, height = 512, width = 512, num_classes = 1000 | 2.008× | batch_size = 64, input_channels = 3, height = 1024, width = 1024, num_classes = 1000 | 2.040× | 4× |

---

> ### Author Response · Authors · 2025-11-21
>
> Weakness 3:
>
> Performance measurement with native pytorch implementation without torch.compile does not reflect a comparison with a true baseline.
>
> Answer 3:
>
> Thank you for bringing up this important point. In our original paper, we followed the convention of most concurrent works [1][2][3], which all evaluate performance with Pytorch Eager (without torch.compile). And we agree that comparing against the PyTorch Eager implementation without torch.compile may not fully reflect the strongest baseline.
>
> In response to this concern, we have conducted additional experiments incorporating torch.compile for the PyTorch reference implementation. We randomly sampled some tasks from three levels(same as the subset  D* mentioned in the original paper line 308-311), and tested the performance with torch.compile,  as shown in Table R2, our method can still achieve an average of 1.1x, 1.53x, 1.14x speedup over baseline for level 1, 2, 3, respectively. Table R3 shows the corresponding results when comparing CudaForge against PyTorch Eager on subset D* .
>
> While the improvements relative to torch.compile (Table R2) are naturally smaller than those relative to PyTorch Eager (Table R3), it shows CudaForge’s strong capability to outperform torch.compile, demonstrating that our performance gains do not rely on Python-level overhead or framework dispatch costs.
>
> To our knowledge, such comparisons against torch.compile are rarely discussed in concurrent work, making this evaluation a meaningful addition. We will include these torch.compile baseline results in the revised version of the paper, together with the corresponding discussion.
>
> **Table R2: Result on ${D}^{*}$, compared with torch.compile**
>
> | Tasks                 | Correct | Performance | Fast1 |
> |-----------------------|---------|-------------|-------|
> | 10 tasks from Level 1 | 100%     | 1.10×    | 50%   |
> | 10 tasks from Level 2 | 100%     | 1.53×    | 80%   |
> | 5 tasks from Level 3  | 100%     | 1.14×    | 40%   |
>
> **Table R3: Result on ${D}^{*}$, compared with PyTorch Eager**
>
> | Tasks                 | Correct | Performance | Fast1 |
> |-----------------------|---------|-------------|-------|
> | 10 tasks from Level 1 | 100%     | 1.49×     | 70%   |
> | 10 tasks from Level 2 | 100%     | 2.15×     | 90%   |
> | 5 tasks from Level 3  | 100%    | 1.56×    | 100%  |
>
> [1] Anne Ouyang, Simon Guo, Simran Arora, Alex L. Zhang, William Hu, Christopher R´e, and Azalia Mirhoseini. Kernelbench: Can llms write efficient gpu kernels?, 2025. URL https://arxiv.org/abs/2502.10517.
>
> [2] Robert Tjarko Lange, Qi Sun, Aaditya Prasad, Maxence Faldor, Yujin Tang, and David Ha. Towards robust agentic cuda kernel benchmarking, verification, and optimization, 2025. URL https://arxiv.org/abs/2509.14279.
>
> [3] Carlo Baronio, Pietro Marsella, Ben Pan, Simon Guo, and Silas Alberti. Kevin: Multi-turn rl for
> generating cuda kernels, 2025. URL https://arxiv.org/abs/2507.11948.

---

> > ### Author Response · Authors · 2025-11-21
> >
> > Question 1:
> >
> > Does CudaForge scale to Triton programming? What performance improvements can be achieved there?
> >
> > Answer 4:
> >
> > Thank you for your insightful comments. We have discussed Triton programming in Answer 1. In brief, the CudaForge workflow itself is language-agnostic, and the two-agent design naturally extends to Triton because both the Coder’s code-generation process and the Judge’s hardware-feedback-based refinement do not rely on CUDA-specific assumptions.
> > As mentioned earlier, we are in the process of extending CudaForge to Triton kernels, and preliminary observations indicate that the framework transfers with minimal modification. If time permits, we will make our best effort to incorporate Triton results in the revised version.
> > Please feel free to let us know if you have any further questions regarding Triton support！
> >
> > Question 2:
> >
> > How does CudaForge methodology evaluate speedup of generated kernels?
> >
> > Answer 5:
> >
> > Thank you for your insightful question. We further clarify the speedup evaluation methodology here (also discussed in Answer 2).
> > For each task, once a kernel passes the correctness checks, we evaluate its performance using the largest input shape among the ten shapes generated for that task. We measure the kernel’s execution time using CUDA events and denote this as $T_{\text{CudaForge}}$. For the baseline, we measure the execution time of the PyTorch reference implementation under the same largest input shape, denoted as $T_{\text{PyTorch}}$. The speedup is then computed as:
> >
> > $$
> > \text{Speedup} = \frac{T_{\text{PyTorch}}}{T_{\text{CudaForge}}}.
> > $$
> >
> >
> > This formulation ensures a fair comparison because (1) both measurements use identical input shapes, and (2) large shapes ensure that GPU computation dominates the runtime rather than framework or OS overhead.
> > Please feel free to let us know if you have any further questions regarding the speedup evaluation. Thank you!
> >
> > Question 3:
> >
> > How does this approach compare against implementations like AlphaEvolve/OpenEvolve?
> >
> > Answer 6:
> >
> > We thank the reviewer for this insightful question! Here we compare AlphaEvolve/OpenEvolve with CudaForge and analyze our advantages.
> >
> > AlphaEvolve and OpenEvolve represent a paradigm of LLM-driven evolutionary search. These systems maintain a population of candidate programs and utilize LLMs as stochastic mutation operators to explore the solution space. The evolution is guided principally by scalar fitness signals (e.g., execution latency) over thousands of generations, operating largely as a "black-box" search process.
> >
> > The most critical distinction lies in the optimization mechanism. Approaches like AlphaEvolve rely on prompt optimization and multi-rounds self-iteration that is largely independent of the specific environmental state and very costly. In contrast, CudaForge employs targeted optimization based on hardware feedback, which explicitly analyzes the current state of the execution environment. This distinction allows CudaForge to rapidly diagnose bottlenecks and converge to superior solutions with significantly higher efficiency.
> >
> > Question 4:
> >
> > How does CudaForge performance metrics look like when compared against torch.compile version of pytorch?
> >
> > Answer 7:
> >
> > Thank you for raising this important question! To address this concern, we conducted additional experiments using the torch.compile version of the PyTorch reference implementation, which is shown in Answer 3 Table R2. These results confirm that CudaForge continues to achieve substantial performance improvements even when the baseline is strengthened with torch.compile.

---

> ### Author Response · Authors · 2025-11-26
>
> Dear Reviewer,
>
> I hope this message finds you well. As the discussion period is nearing its end with less than one week remaining, l want to ensure we have addressed all your concerns satisfactorily. If there are any additional points or feedbacks you'd like us to consider, please let us know. Your insights are invaluable to us, and we are eager to address any remaining issues to improve our work.
>
> Thank you for your time and effort in reviewing our paper!

---

### Official Review · Reviewer_TYZF · 2025-11-01

**Soundness:** 3
**Presentation:** 3
**Contribution:** 2
**Rating:** 4
**Confidence:** 3

**Summary:**

CudaForge proposed two agents: Coder and Judge. The Judge analyzes runtime errors and nsight compute performance metrics, identifies bottlenecks such as memory stalls or low occupancy. It provides optimization feedback to the Coder. Coder regenerates an improved kernel. This loop continues until convergence. CudaForge does not requrie reinforcement learning or training.

**Strengths:**

1. Originality
Incorporating nsight compute profiling data is an effective approach. The outcome is highly verifiable. This bridges a gap between abstract code generation and hardware-aware tuning, mimicking expert workflows in a systematic way. The separation of roles into a Coder and Judge are sound and understoodable. CudaForge performs optimization purely at inference time, showing that meaningful performance gains are achievable without learning-based fine-tuning.

2. Quality
The evaluation covered tasks from KernelBench, multiple difficulty levels, and A100 / RTX 6000 / 4090 / 3090 setup.
The comparisons against OpenAI-o3, Kevin are fair. Ablations isolate the impact of correction vs. optimization feedback, demonstrating sound causal reasoning behind design choices. The authors report correctness, speedup and detailed prompts. It is helpful for other researchers to reproduce the result.

3. Clarity
human vs. agent workflow diagram effectively illustrate the iterative process. The case study of CrossEntropyLoss optimization makes the workflow intuitively understoodable. The breakdown of Judge behavior, metric selection algorithms, and prompt templates offers transparency.

4. Significance
The work may establish a new design that uses nsight compute profiling data from real hardware. This is applicable beyond applicable beyond CUDA. Automating CUDA kernel optimization directly addresses one of the most labor-intensive bottlenecks in deep learning systems development. The framework could democratize performance engineering and lower the barrier for custom GPU optimization. Most importantly, it's training free.

**Weaknesses:**

the overall multi-agent refinement structure follows a familiar template used in prior agent-based code generation frameworks using  self-refine. The core advance lies in the feedback modality rather than a fundamentally new learning or reasoning principle. The paper draws a hard line between training-free and RL-based paradigms but doesn’t explore hybrid approaches. In practice, the line can be blurred. If the kernel perf is verifable, it's possible to train the model for better answers

While the paper reports end-to-end time (≈25 min per kernel), it doesn’t dissect where that time is spent — compilation, profiling, or LLM inference. This makes it difficult to assess which parts of the workflow dominate cost and how well it scales for large codebases.

**Questions:**

How do we know the kernel is 100% numerically correct? It's hard to just run a few input and claim it's right, because the input are limited samples.

If perf and correctness are verifiable, is it possible to extend to RL style training to improve the model

---

> ### Author Response · Authors · 2025-11-21
>
> Thanks for your time and effort in reviewing our work. We sincerely appreciate your recognition of our contributions in effective training-free framework, NCU incorporation,comprehensive evaluation, case study and practical value. Also we thank your valuable suggestions for our paper. Below, we address your concerns point by point and we’ll revise our paper according to your suggestions. We would appreciate it if you could let us know whether your concerns are addressed by our response.
>
> Question 1:
>
> the overall multi-agent refinement structure follows a familiar template used in prior agent-based code generation frameworks using self-refine. The core advance lies in the feedback modality rather than a fundamentally new learning or reasoning principle. The paper draws a hard line between training-free and RL-based paradigms but doesn’t explore hybrid approaches. In practice, the line can be blurred. If the kernel perf is verifiable, it's possible to train the model for better answers
>
> Answer 1:
>
> Thank you for this thoughtful comment. We would like to clarify how CudaForge differs from prior self-refine frameworks and why the design choice of a training-free, hardware-feedback-driven workflow is central to the contribution of the paper, and we would like to discuss the relationship between CudaForge and RL-based paradigms:
>
> **Leverage hardware feedback**. One of the core advances is leveraging hardware feedback. While CudaForge adopts an iterative refinement workflow similar in form to prior self-refine frameworks, the central contribution of our approach lies not in the iterative structure itself but in the feedback modality and optimization process. Unlike previous agent-based code generation methods [1,2,3], which mainly rely on static textual feedback or syntax-level error messages, CudaForge introduces a hardware-aware Judge that interprets detailed Nsight Compute profiling data, such as memory stall breakdowns, warp occupancy, cache statistics, register usage, and leverages these execution metrics together with GPU specifications to guide Coder’s kernel optimization.
>
> **Systematic Integration of Hardware Feedback**. Beyond merely accessing hardware feedback, another core advance is proposing a systematic methodology for integrating hardware feedback into an agentic workflow. Our methodology selects a high-quality subset of NCU metrics, which are highly related to the kernel’s performance and represent critical bottlenecks. It enables our multi-agent system to focus on addressing only one critical program bottleneck in each round, and eventually optimize overall kernel performance step by step in iterative rounds, just like human experts’ real workflow.
>
> **Separation of Coder and Judge**. Moreover, separating Coder’s and Judge’s roles is also a key advance. The separation between the Coder and Judge is designed specifically to make hardware feedback interpretable for the model. By assigning the Coder solely the task of generation and the Judge the role of evaluating correctness and performance using hardware and runtime information, CudaForge effectively distributes the “cognitive” load between two specialized agents. This division of labor mirrors human expert workflows and reduces the risk of overlooking errors or performance inefficiencies, constituting a methodological contribution that extends beyond the general self-refinement template.
>
> Regarding the distinction between training-free and RL-based paradigms: our intention is not to draw a strict ideological boundary, but rather to highlight that CudaForge achieves substantial correctness and speedup gains without any additional training or reinforcement learning, thereby demonstrating that LLMs already possess sufficient latent capability to perform hardware-aware kernel optimization when provided with appropriately structured feedback. This offers a practical advantage: the framework can be applied immediately to new GPUs, new models, and new tasks without retraining costs.
>
> We agree that hybrid approaches (e.g., using verified kernels as reinforcement signals) are possible, and we view them as a promising direction. However, this lies outside the scope of the current work, whose goal is explicitly to show that training-free hardware-feedback-driven refinement is already effective, reproducible, and broadly applicable. The strong empirical results across multiple GPUs and models demonstrate that the contribution does not rely on training but on the novel use of profiler feedback within the agentic refinement loop.
>
> Again we highly appreciate your insightful comments!
>
> [1] Self-Refine: Iterative Refinement with Self-Feedback. url={https://arxiv.org/abs/2303.17651}
>
> [2] Carlo Baronio, Pietro Marsella, Ben Pan, Simon Guo, and Silas Alberti. Kevin: Multi-turn rl for generating cuda kernels, 2025. URL https://arxiv.org/abs/2507.11948.
>
> [3] Reflexion: Language Agents with Verbal Reinforcement Learning. url={https://arxiv.org/abs/2303.11366}

---

> > ### Author Response · Authors · 2025-11-21
> >
> > Question 2:
> >
> > While the paper reports end-to-end time (≈25 min per kernel), it doesn’t dissect where that time is spent — compilation, profiling, or LLM inference. This makes it difficult to assess which parts of the workflow dominate cost and how well it scales for large codebases.
> >
> > Answer 2:
> >
> > Thank you for raising this concern. Where the 25 mins is spent is shown in Table R1. For a typical KernelBench task with ten refinement rounds, the ≈25-minute end-to-end runtime is dominated by Nsight Compute profiling, which takes approximately 10–12 minutes in total. Kernel compilation accounts for roughly 2–3 minutes, while LLM inference contributes about 9–11 minutes across all rounds. This breakdown shows that the overall runtime is determined primarily by profiling rather than model latency or compilation overhead. Since kernels are independent and can be processed concurrently, CudaForge scales effectively to larger codebases, with throughput largely governed by the degree of parallelism available for profiling rather than limitations of the refinement framework itself.
> >
> > | Category                 | Time          |
> > |--------------------------|---------------|
> > | Nsight Compute profiling | 10–12 minutes |
> > | LLM inference            | 9–11 minutes  |
> > | Kernel compilation       | 2–3 minutes   |
> >
> >
> > Question 3:
> >
> > How do we know the kernel is 100% numerically correct? It's hard to just run a few input and claim it's right, because the input are limited samples.
> >
> > Answer 3:
> >
> > Thank you for raising this important question. To mitigate the risk associated with limited input samples, we explicitly evaluate each candidate kernel on multiple input shapes rather than relying on a single test case. This design is intended to stress the kernel under diverse tensor sizes and reduce the chance that a kernel passes correctness checks only for a narrowly specified input.
> >
> > Concretely, CudaForge tests correctness using ten diverse input shapes, each paired with randomly generated tensors. A kernel is accepted as correct only if it compiles successfully and its outputs match those of the PyTorch reference implementation within a tolerance of 1e−4 on all ten cases. This multi-shape, multi-input procedure significantly decreases the likelihood of accidental correctness and allows us to approximate functional correctness as reliably as possible in practice.
> >
> > We would also like to emphasize that this evaluation protocol (using test cases) follows the established standard used across existing GPU kernel benchmarks and all concurrent LLM-based CUDA generation works [1–4]. CudaForge adheres to and strengthens this standard. We will make this clearer in the revised version.
> >
> > [1] Anne Ouyang, Simon Guo, Simran Arora, Alex L. Zhang, William Hu, Christopher R´e, and Azalia Mirhoseini. Kernelbench: Can llms write efficient gpu kernels?, 2025. URL https://arxiv.org/abs/2502.10517.
> >
> > [2] Robert Tjarko Lange, Qi Sun, Aaditya Prasad, Maxence Faldor, Yujin Tang, and David Ha. Towards robust agentic cuda kernel benchmarking, verification, and optimization, 2025. URL https://arxiv.org/abs/2509.14279.
> >
> > [3] Carlo Baronio, Pietro Marsella, Ben Pan, Simon Guo, and Silas Alberti. Kevin: Multi-turn rl for generating cuda kernels, 2025. URL https://arxiv.org/abs/2507.11948.
> >
> > [4] DeepReinforce Team. Cuda-l1: Improving cuda optimization via contrastive reinforcement learning. arXiv preprint arXiv:2507.14111, 2025.

---

> > > ### Author Response · Authors · 2025-11-21
> > >
> > > Question 4:
> > >
> > > If perf and correctness are verifiable, is it possible to extend to RL style training to improve the model
> > >
> > > Answer 4:
> > >
> > > Thank you for the insightful question. Here we would like to clarify the focus of our work and discuss the possibility of RL training. In principle, once correctness and performance can be reliably verified, it is indeed possible to incorporate reinforcement learning signals to further improve the underlying model. We agree that RL-based approaches represent a promising direction for leveraging verifiable kernel feedback.
> > >
> > > However, the focus of our work is different: CudaForge is designed to demonstrate that a training-free framework, when equipped with fine-grained hardware feedback and an iterative Coder/Judge workflow, can already achieve strong correctness and performance without any additional model tuning. Our contribution lies in showing that existing LLMs possess sufficient latent capability for hardware-guided kernel optimization when provided with the right feedback structure, and that this approach is practical, reproducible, and immediately deployable across GPUs and tasks without further training cost. Of course, we agree with the reviewer that performing RL to further improve the model is very useful, and this will be orthogonal to our development.
> > >
> > > Let us provide two perspectives regarding RL vs our agent workflow.
> > >
> > > 1. We can actually provide some interesting insights between purely workflow based algorithm and RL based algorithm.  Indeed, we compare CudaForge initialized with QwQ-32B as Coder and OpenAI-o3 as Judge (line 445 in the paper) against Kevin-32B, an RL-enhanced model trained from QwQ-32B (line 333). As shown in Table R2, CudaForge (QwQ + o3) outperforms Kevin-32B, despite being entirely training-free. This suggests that a strong workflow can surpass RL methods even before applying any training.
> > >
> > > | Method                     | Correct | Performance | $Fast_1$ |
> > > |----------------------------|---------|-------------|-------|
> > > | **CudaForge (QwQ+o3)**     | 84%     | 0.790x       | 44%   |
> > > | **Kevin-32B**              | 64%     | 0.608x       | 36%   |
> > >
> > >
> > > 2. Effective RL requires a reliable and efficient environment, and building such an environment is often the true bottleneck. Before conducting RL, one must ensure that correctness checks, performance evaluation and the iterative structure are well-designed and stable. CudaForge provides precisely such a workflow, and we believe it forms a solid foundation on top of which RL methods can later be explored.
> > >
> > > We view RL as a complementary and promising extension to CudaForge. It is still a very new paradigm in LLM-based CUDA optimization, and many fundamental questions remain open, including how to formalize the model’s interaction with the environment, how to reliably verify kernel performance, and how to design stable and meaningful reward signals. These challenges require substantial additional research. Our focus in this work is to demonstrate that a training-free, profiler-guided multi-agent framework is already highly effective, but we fully agree that RL represents an interesting future direction. Thank you again for your insightful comments.

---

> ### Author Response · Authors · 2025-11-26
>
> Dear Reviewer,
>
> I hope this message finds you well. As the discussion period is nearing its end with less than one week remaining, l want to ensure we have addressed all your concerns satisfactorily. If there are any additional points or feedbacks you'd like us to consider, please let us know. Your insights are invaluable to us, and we are eager to address any remaining issues to improve our work.
>
> Thank you for your time and effort in reviewing our paper!

---

### Official Review · Reviewer_wyaT · 2025-11-02

**Soundness:** 2
**Presentation:** 3
**Contribution:** 2
**Rating:** 2
**Confidence:** 3

**Summary:**

The paper proposes CudaForge, a framework for LLM-based kernel generation that incorporates separate Coder and Judge models, with the judge being guided by actual hardware profiling feedback.

**Strengths:**

The inclusion of hardware execution feedback seems like a critical step in improving autogenerated kernels, and the investigation into which metrics actually matter as feedback is generally useful beyond this paper (not only to limit the context to be put into the LLM, but also, gathering reduced statistics makes profiling faster).

The method achieves very good correctness scores, while being computationally cheaper than competitors; in particular, it is training-free.

Having an example of how the LLM achieves its improvements and how it fails in certain cases  (appendix A) is very good; though I'd like to  see the actual kernel implementations, to verify whether the improvements proposed in figure 4 are actually implemented.

Also, thanks for including the actual prompts used in this work; this should make it more easily reproducable than some of its competitors.

**Weaknesses:**

Unfortunately, KernelBench, as a benchmark, is quite flawed, because many of its tasks use shapes that are too small, exacerbating the overheads induced by not using torch.compile as the baseline.

It seems hard to believe that on something as essential as cross-entropy, there'd be a 4x speedup left on the table;

Figure 4 suggests that the framework is doing something promising, but I am very skeptical about the reported speedups reflecting meaningful scenarios.

I'm not sure the "Comparison with O3-10-O (optimization-only Judge)" paragraph adds anything substantial to the paper; this might be better placed in the appendix, and Figure 4 moved to the main part.

**Questions:**

How does the model perform on larger input shapes. Have you independently verified that the model is not exploiting some weakness in the evaluation procedure, especially in cases where very large speed-ups are reported.

What fraction of speed-of-light is achieved by the baseline, what by the generated kernels.

---

> ### Author Response · Authors · 2025-11-21
>
> Thanks for your time and effort in reviewing our work. We sincerely appreciate your recognition of our contributions in hardware feedback design, effective training-free framework, case study and efforts for reproductivity. Also we thank your valuable suggestions for our paper. Below, we address your concerns point by point and we’ll revise our paper according to your suggestions. We would appreciate it if you could let us know whether your concerns are addressed by our response.
>
> Weakness 1:
>
> Unfortunately, KernelBench, as a benchmark, is quite flawed, because many of its tasks use shapes that are too small, exacerbating the overheads induced by not using torch.compile as the baseline.
>
> Answer 1:
>
> Thank you for raising this important concern regarding the limitations of KernelBench.
>
> First, we would like to clarify that KernelBench is nearly the **only benchmark for LLM CUDA generation** by the deadline of ICLR submission, except the robust-kbench [2] published on 16th, September, 8 days before the deadline. For fair comparison with state-of-the-arts, KernelBench therefore remained the only established and practically usable benchmark within the submission timeframe. As a result, KernelBench remains the standard evaluation protocol used not only by our work but also by **nearly all concurrent submissions** in this area [1,3,4,5,6]. Thus, using KernelBench is consistent with the prevailing evaluation practice for this emerging research direction.
>
> Second, we fully acknowledge the statement you pointed out: some KernelBench tasks indeed involve relatively small input shapes, which may make baseline performance more susceptible to Python-level overhead. This is a genuine limitation of the benchmark. Although the page limit prevented us from providing a detailed explanation in the main text before the full paper deadline, we have taken concrete steps in the original experiments to mitigate this issue during our experimental design:
>
> Specifically, we do not directly use the single input shape provided by KernelBench. For each task, we query the GPT-4o API to generate ten diverse input shapes, ranging from moderately sized examples (e.g., 4096×4096) to shapes that utilize a substantial portion of the RTX 6000 GPU’s memory capacity(e.g., 16384×16384), as shown in Table R1.
>
> Then, for **Correctness Evaluation** : a kernel is deemed correct only if it passes all ten input shapes.
> For **Kernel Optimization**, we profile the kernel using the largest input shape, and the Judge agent performs optimization based on hardware feedback from this large-shape profile.
>
> After N rounds of iteration, we will choose the most efficient correct kernel as the final solution. Finally, when computing speedup relative to the PyTorch baseline, we always compare execution times using the largest input shape. This design serves two purposes: **(1) Large shapes ensure GPU computation, not PyTorch framework overhead or OS operations, dominates runtime, thereby providing a fairer and more meaningful comparison for kernel optimization. (2) Optimizing CUDA kernels is fundamentally motivated by large-scale computation, such as the matrix operations frequently encountered in LLM-level workloads.** Thus, evaluating performance on the largest shape aligns with the real-world goals of CUDA kernel optimization.
>
> Thirdly, thank you for your insightful feedback to include torch.compile(). According to the limitation of computing resources, we didn’t run experiments based on torch.compile(). In our original paper, we followed the convention of **most of concurrent works** [1,2,3,5], which all evaluate performance with Pytorch Eager (without torch.compile). And we agree that comparing against the PyTorch Eager implementation without torch.compile may not fully reflect the strongest baseline.
>
> In response to this concern, we have conducted additional experiments incorporating torch.compile for the PyTorch reference implementation. We randomly sampled some tasks from three levels(same as the subset  D* mentioned in the original paper line 308-311), and tested the performance with torch.compile,  as shown in Table R2, our method can still achieve an average of 1.1x, 1.53x, 1.14x speedup over baseline for level 1, 2, 3, respectively. Table R3 shows the corresponding results when comparing CudaForge against PyTorch Eager on subset D* .
>
> While the improvements relative to torch.compile (Table R2) are naturally smaller than those relative to PyTorch Eager (Table R3), it shows CudaForge’s strong capability to outperform torch.compile, demonstrating that our performance gains do not rely on Python-level overhead or framework dispatch costs.
>
> To our knowledge, such comparisons against torch.compile are rarely discussed in concurrent work, making this evaluation a meaningful addition. We will include these torch.compile baseline results in the revised version of the paper, together with the corresponding discussion.

---

> > ### Author Response · Authors · 2025-11-21
> >
> > Answer 1:
> >
> > **Table R1. Performance under KernelBench input sizes and maximum test sizes**
> >
> > | Task            | Size in KernelBench                                      | Perf     | Max Size in Test                                      | Perf     | Change in Size |
> > |-----------------|-----------------------------------------------------------|----------|--------------------------------------------------------|----------|----------------|
> > | Level 1 Task 8  | M = 8205, N = 2949, K = 5921 | 0.996× | M = 32820, N = 11796, K = 23684 | 0.993× | 64× |
> > | Level 1 Task 15 | M = 4096, N = 4096 | 3.063× | M = 16384, N = 16384 | 3.385× | 16× |
> > | Level 2 Task 21 | batch_size = 128, in_channels = 8, out_channels = 32, height = width = 256, kernel_size = 3, num_groups = 8 | 1.531× | batch_size = 128, in_channels = 8, out_channels = 32, height = width = 512, kernel_size = 3, num_groups = 8 | 1.449× | 4× |
> > | Level 2 Task 75 | batch_size = 1024, in_features = 8192, out_features = 8192, num_groups = 512 | 1.030× | batch_size = 1024, in_features = 19384, out_features = 19384, num_groups = 1024 | 1.017× | 8× |
> > | Level 3 Task 18 | batch_size = 64, input_channels = 3, height = 512, width = 512, num_classes = 1000 | 2.008× | batch_size = 64, input_channels = 3, height = 1024, width = 1024, num_classes = 1000 | 2.040× | 4× |
> >
> > **Table R2: Result on ${D}^{*}$, compared with torch.compile**
> >
> > | Tasks                 | Correct | Performance | Fast1 |
> > |-----------------------|---------|-------------|-------|
> > | 10 tasks from Level 1 | 100%     | 1.10×    | 50%   |
> > | 10 tasks from Level 2 | 100%     | 1.53×    | 80%   |
> > | 5 tasks from Level 3  | 100%     | 1.14×    | 40%   |
> >
> > **Table R3: Result on ${D}^{*}$, compared with PyTorch Eager**
> >
> > | Tasks                 | Correct | Performance | Fast1 |
> > |-----------------------|---------|-------------|-------|
> > | 10 tasks from Level 1 | 100%     | 1.49×     | 70%   |
> > | 10 tasks from Level 2 | 100%     | 2.15×     | 90%   |
> > | 5 tasks from Level 3  | 100%    | 1.56×    | 100%  |
> >
> > [1] Anne Ouyang, Simon Guo, Simran Arora, Alex L. Zhang, William Hu, Christopher R´e, and Azalia Mirhoseini. Kernelbench: Can llms write efficient gpu kernels?, 2025. URL https://arxiv.org/abs/2502.10517.
> >
> > [2] Robert Tjarko Lange, Qi Sun, Aaditya Prasad, Maxence Faldor, Yujin Tang, and David Ha. Towards robust agentic cuda kernel benchmarking, verification, and optimization, 2025. URL https://arxiv.org/abs/2509.14279.
> >
> > [3] Carlo Baronio, Pietro Marsella, Ben Pan, Simon Guo, and Silas Alberti. Kevin: Multi-turn rl for generating cuda kernels, 2025. URL https://arxiv.org/abs/2507.11948.
> >
> > [4] DeepReinforce Team. Cuda-l1: Improving cuda optimization via contrastive reinforcement learning. arXiv preprint arXiv:2507.14111, 2025.
> >
> > [5] EvoEngineer: Mastering Automated CUDA Kernel Code Evolution with Large Language Models,url {https://openreview.net/forum?id=LU27DiW5ik}
> >
> > [6] STARK: Strategic Team of Agents for Refining Kernels,url={https://openreview.net/forum?id=nWaZTH1JMx}

---

> ### Author Response · Authors · 2025-11-21
>
> Weakness 2:
>
> It seems hard to believe that on something as essential as cross-entropy, there'd be a 4x speedup left on the table; Figure 4 suggests that the framework is doing something promising, but I am very skeptical about the reported speedups reflecting meaningful scenarios.
>
> Answer 2:
>
> Thank you for raising concerns about the effectiveness of cross-entropy’s speedup. Here we demonstrate the actual kernel implementations of KernelBench Level1 Task95 (Cross Entropy).
> Our framework used three kinds of optimization methods in the final implementation in this case.
>
> First, improving warp-level parallelism, including assigning all classes of one sample to one warp, using warp shuffle for max/sum reduce to avoid accessing shared/global memory for this, and using __forceinline__ + **#pragma unroll** to reduce branches.
>
> Second, better memory access pattern, achieved by vectorized and coalesced data loading with float4. Third, two different pipelining designs for small input size and big input size. When the input size is small, using shared memory as much as possible, specifically, load logits into shared memory and overlap this load with the computing of max, then the next read can just read shared memory without accessing global memory. While when input size is larger and chunk size far exceeds the size of shared memory, leading to higher smem occupancy and bank conflicts in smem becoming the bottleneck, directly read data from global memory and build a pipeline with double buffers, overlapping memory loads for chunk N+1 with arithmetic on chunk N.
>
> Lastly, our agents even did the kernel fusion of softmax and reduce, significantly reducing the overhead.
>
> This also proves that **our agents will consider different input sizes and give specific customized optimization for different input sizes!** And we also want to emphasize that the input size is given by the user, and our agent system can **handle various situations and customize a high-performance kernel** for users, that is the core value of the agentic cuda programming framework.
>
> Finally, why is the performance of the default pytorch implementation so bad? We think it is because cross_entropy is an old but not popular kernel and somehow forgotten by developers in our community, leaving it unoptimized for too long time.
>
> We also attached the cuda codes generated by our framework below. We have to admit that it might not be perfect, but it was **totally automatically generated by agents** and has better performance than baseline.
>
> ```cpp
> #include <torch/extension.h>
> #include <ATen/cuda/CUDAContext.h>
> #include <cuda.h>
> #include <cuda_runtime.h>
> #include <float.h>
>
> #define CUDA_CHECK_ERRORS()
>   do {
>     cudaError_t err = cudaGetLastError();
>     TORCH_CHECK(err == cudaSuccess, "CUDA kernel failed : ",
>                 cudaGetErrorString(err));
>   } while (0)
>
> // ---------------------------------------------------------------------
> // Warp-level helpers
> // ---------------------------------------------------------------------
> template <typename T>
> __device__ __forceinline__ T warp_max(T v) {
>     #pragma unroll
>     for (int off = 16; off > 0; off >>= 1)
>         v = max(v, __shfl_down_sync(0xffffffff, v, off));
>     return v;
> }
>
> template <typename T>
> __device__ __forceinline__ T warp_sum(T v) {
>     #pragma unroll
>     for (int off = 16; off > 0; off >>= 1)
>         v += __shfl_down_sync(0xffffffff, v, off);
>     return v;
> }
>
> // ---------------------------------------------------------------------
> // Kernel configuration
> // ---------------------------------------------------------------------
> #define WARPS_PER_BLOCK 4
> #define THREADS_PER_BLOCK (WARPS_PER_BLOCK * 32)
>
> // ---------------------------------------------------------------------
> // Vectorised register-only kernel
> // ---------------------------------------------------------------------
> __global__ void fused_cross_entropy_kernel_regs_v3(
>         const float*  __restrict__ logits,
>         const int64_t* __restrict__ targets,
>         float*        __restrict__ losses,
>         const int num_classes,
>         const int total_samples) {
>
>     const int lane            = threadIdx.x & 31;
>     const int warp_in_block   = threadIdx.x >> 5;
>     const int warp_global     = blockIdx.x * (blockDim.x >> 5) + warp_in_block;
>     const int sample          = warp_global;
>
>     if (sample >= total_samples) return;
>
>     const float* sample_logits = logits + sample * num_classes;
>     const int64_t tgt          = targets[sample];

---

> > ### Author Response · Authors · 2025-11-21
> >
> > Answer 2：
> >
> > ```cpp
> >
> > // -----------------------------------------------------------------
> >     // Pass 2 : Σexp & pick target logit  (software-pipelined 2-buffer)
> >     // -----------------------------------------------------------------
> >     float local_sum  = 0.f;
> >     float tgt_logit  = 0.f;
> >
> >     // pre-load first segment
> >     int base = lane * 4;
> >     float4 buf_cur, buf_next;
> >     bool   have_next = (base < vec_limit);
> >     if (have_next) buf_cur = *(const float4*)(sample_logits + base);
> >
> >     base += 128;
> >     while (have_next) {
> >         have_next = (base < vec_limit);
> >         if (have_next)
> >             buf_next = *(const float4*)(sample_logits + base);
> >
> >         // compute on buf_cur
> >         float vals[4] = {buf_cur.x, buf_cur.y, buf_cur.z, buf_cur.w};
> >         #pragma unroll
> >         for (int k = 0; k < 4; ++k) {
> >             int idx = (base - 128) + k;    // idx of buf_cur element
> >             if (idx < num_classes) {
> >                 float l = vals[k];
> >                 local_sum += __expf(l - warp_max_val);
> >                 if (idx == tgt) tgt_logit = l;
> >             }
> >         }
> >
> >         buf_cur = buf_next;
> >         base   += 128;
> >     }
> >
> >     // tail (≤3)
> >     for (int idx = vec_limit + lane; idx < num_classes; idx += 32) {
> >         float l = sample_logits[idx];
> >         local_sum += __expf(l - warp_max_val);
> >         if (idx == tgt) tgt_logit = l;
> >     }
> >
> >     float warp_sum_val = warp_sum(local_sum);
> >     warp_sum_val       = __shfl_sync(0xffffffff, warp_sum_val, 0);
> >
> >     float warp_tgt_log = warp_sum(tgt_logit);
> >     warp_tgt_log       = __shfl_sync(0xffffffff, warp_tgt_log, 0);
> >
> >     // -----------------------------------------------------------------
> >     // Write result
> >     // -----------------------------------------------------------------
> >     if (lane == 0) {
> >         float log_p = warp_tgt_log - warp_max_val - logf(warp_sum_val);
> >         losses[sample] = -log_p;
> >     }
> > }
> >
> > // ---------------------------------------------------------------------
> > // Shared-memory buffered kernel (vectorised load into smem)
> > // ---------------------------------------------------------------------
> > __global__ void fused_cross_entropy_kernel_smem_v2(
> >         const float*  __restrict__ logits,
> >         const int64_t* __restrict__ targets,
> >         float*        __restrict__ losses,
> >         const int num_classes,
> >         const int total_samples) {
> >
> >     extern __shared__ float smem[];                // WARPS_PER_BLOCK * num_classes
> >     const int lane          = threadIdx.x & 31;
> >     const int warp_in_block = threadIdx.x >> 5;
> >     const int warp_global   = blockIdx.x * (blockDim.x >> 5) + warp_in_block;
> >     const int sample        = warp_global;
> >
> >     if (sample >= total_samples) return;
> >
> >     const float* sample_logits = logits + sample * num_classes;
> >     float* warp_smem = smem + warp_in_block * num_classes;
> >     const int64_t tgt = targets[sample];
> >
> >     // ---------------------------------------------------------------
> >     // 1. Load logits -> shared (float4), compute local max
> >     // ---------------------------------------------------------------
> >     float local_max = -FLT_MAX;
> >     const int vec_limit = (num_classes & ~3);
> >
> >     for (int base = lane * 4; base < vec_limit; base += 128) {
> >         float4 v = *(const float4*)(sample_logits + base);
> >         warp_smem[base + 0] = v.x;
> >         warp_smem[base + 1] = v.y;
> >         warp_smem[base + 2] = v.z;
> >         warp_smem[base + 3] = v.w;
> >
> >         local_max = fmaxf(local_max, v.x);
> >         local_max = fmaxf(local_max, v.y);
> >         local_max = fmaxf(local_max, v.z);
> >         local_max = fmaxf(local_max, v.w);
> >     }
> >     // tail
> >     for (int idx = vec_limit + lane; idx < num_classes; idx += 32) {
> >         float l = sample_logits[idx];
> >         warp_smem[idx] = l;
> >         local_max = fmaxf(local_max, l);
> >     }
> >
> >     float warp_max_val = warp_max(local_max);
> >     warp_max_val       = __shfl_sync(0xffffffff, warp_max_val, 0);
> >
> >     // ---------------------------------------------------------------
> >     // 2. Σexp & pick target using cached logits
> >     // ---------------------------------------------------------------
> >     float local_sum = 0.f;
> >     float tgt_logit = 0.f;
> >
> >     for (int idx = lane; idx < num_classes; idx += 32) {
> >         float l = warp_smem[idx];
> >         local_sum += __expf(l - warp_max_val);
> >         if (idx == tgt) tgt_logit = l;
> >     }
> >
> >     float warp_sum_val = warp_sum(local_sum);
> >     warp_sum_val       = __shfl_sync(0xffffffff, warp_sum_val, 0);
> >
> >     float warp_tgt_log = warp_sum(tgt_logit);
> >     warp_tgt_log       = __shfl_sync(0xffffffff, warp_tgt_log, 0);
> >
> >     // ---------------------------------------------------------------
> >     // 3. Write loss
> >     // ---------------------------------------------------------------
> >     if (lane == 0) {
> >         float log_p = warp_tgt_log - warp_max_val - logf(warp_sum_val);
> >         losses[sample] = -log_p;
> >     }
> > }

---

> > > ### Author Response · Authors · 2025-11-21
> > >
> > > Answer 2:
> > >
> > > ```cpp
> > > // ---------------------------------------------------------------------
> > > // Wrapper – identical public API
> > > // ---------------------------------------------------------------------
> > > torch::Tensor fused_cross_entropy_cuda(torch::Tensor logits,
> > >                                        torch::Tensor targets) {
> > >     TORCH_CHECK(logits.is_cuda(),  "logits must be a CUDA tensor");
> > >     TORCH_CHECK(targets.is_cuda(), "targets must be a CUDA tensor");
> > >     TORCH_CHECK(logits.dtype()  == torch::kFloat32, "logits must be float32");
> > >     TORCH_CHECK(targets.dtype() == torch::kInt64,   "targets must be int64");
> > >     TORCH_CHECK(logits.dim() == 2,  "logits must be {B, C}");
> > >     TORCH_CHECK(targets.dim() == 1, "targets must be {B}");
> > >     TORCH_CHECK(logits.size(0) == targets.size(0),
> > >                 "Batch size mismatch between logits and targets");
> > >
> > >     const int B = logits.size(0);
> > >     const int C = logits.size(1);
> > >
> > >     const int blocks = (B + WARPS_PER_BLOCK - 1) / WARPS_PER_BLOCK;
> > >     const size_t smem_bytes = static_cast<size_t>(C) * WARPS_PER_BLOCK * sizeof(float);
> > >     const bool use_smem = (smem_bytes <= 64 * 1024);
> > >
> > >     auto losses = torch::empty({B}, logits.options());
> > >     cudaStream_t stream = at::cuda::getCurrentCUDAStream();
> > >
> > >     if (use_smem) {
> > >         fused_cross_entropy_kernel_smem_v2<<<blocks, THREADS_PER_BLOCK,
> > >                                              smem_bytes, stream>>>(
> > >             logits.data_ptr<float>(),
> > >             targets.data_ptr<int64_t>(),
> > >             losses.data_ptr<float>(),
> > >             C, B);
> > >     } else {
> > >         fused_cross_entropy_kernel_regs_v3<<<blocks, THREADS_PER_BLOCK,
> > >                                              0, stream>>>(
> > >             logits.data_ptr<float>(),
> > >             targets.data_ptr<int64_t>(),
> > >             losses.data_ptr<float>(),
> > >             C, B);
> > >     }
> > >
> > >     CUDA_CHECK_ERRORS();
> > >     return losses.mean();
> > > }

---

> > > > ### Author Response · Authors · 2025-11-21
> > > >
> > > > Weakness 3:
> > > >
> > > > I'm not sure the "Comparison with O3-10-O (optimization-only Judge)" paragraph adds anything substantial to the paper; this might be better placed in the appendix, and Figure 4 moved to the main part.
> > > >
> > > > Answer 3:
> > > >
> > > > Thank you for your insightful suggestions about our paper writing. We will revise it in the new version, where we will add numerical results and polish our sentences. For Figure 4, we moved it from the main part to the appendix before ICLR submission, because of limited pages. But your suggestion also makes sense. We will try to polish our paper again for better readability and presentation.
> > > >
> > > >
> > > > Question 1:
> > > >
> > > > How does the model perform on larger input shapes? Have you independently verified that the model is not exploiting some weakness in the evaluation procedure, especially in cases where very large speed-ups are reported.
> > > >
> > > > Answer 4:
> > > >
> > > > Thank you very much for your insightful comments. For input shapes, as discussed in **Answer 1**, our evaluation is explicitly designed to avoid relying on small shapes(e.g. 4096*4096).
> > > >
> > > > For verifying evaluation procedure, as mentioned in line 470-475 in the original paper, we manually checked evaluation procedure to avoid hacking behaviour. Thus, CudaForge’s performance reported in the paper is valid. We would also like to emphasize that this manual inspection does not compromise the automation of CudaForge; In practice, people only need to perform it once.
> > > >
> > > > You can also refer the previous comments, where we provide code of KernelBench Level1 Task 95. This may help us to demonstrate results provided by our agent workflow.
> > > >
> > > >
> > > >
> > > > Question 2:
> > > >
> > > > What fraction of speed-of-light is achieved by the baseline, what by the generated kernels.
> > > >
> > > > Answer 5:
> > > >
> > > > Please let us still take the cross_entropy as an example here. In this case, the baseline includes 2 kernels, named nll_loss_forward_reduce_cuda_kernel_2d and softmax_warp_forward, respectively. Part of SOL of these two kernels are shown in below:
> > > >
> > > > **Table R4: baseline kernels: reduce & softmax**
> > > > | Kernel   | DRAM Freq | SM Freq | Mem Throughput | DRAM Throughput | Computer(SM) Throughput |
> > > > |----------|-----------|---------|-----------------|------------------|--------------------------|
> > > > | reduce   | 6.46      | 1.45    | 0.47%           | 0.47%            | 0.01%                    |
> > > > | softmax  | 5.4       | 1.17    | 83.41%          | 83.41%           | 22.52                    |
> > > >
> > > > And according to our profiling, the reduce kernel seriously bottleneck the program due to extremely low resource utilization.
> > > >
> > > > But in the cuda code generated by our agent systems, due to kernel fusion and various optimization methods as we shown in **Answer 2**, the SOL is much better as shown in below:
> > > >
> > > > **Table R5: ours fused kernel**
> > > > | Kernel | DRAM Freq | SM Freq | Mem Throughput | DRAM Throughput | Computer(SM) Throughput |
> > > > |--------|-----------|---------|-----------------|------------------|--------------------------|
> > > > | ours   | 5.93      | 1.29    | 66.67%          | 66.67%           | 30.99%                   |
> > > >
> > > > Beyond this case, we have observed many similar scenarios in other cases. And we also want to mention that even when the torch.compile successfully fused default kernels, there will be extra overhead incurred by the pytorch compiler. But our agent system can **generate a “manually fused” kernel without extra compiler overhead**. This also explains why the performance of the generated kernel is better.

---

> ### Author Response · Authors · 2025-11-26
>
> Dear Reviewer,
>
> I hope this message finds you well. As the discussion period is nearing its end with less than one week remaining, l want to ensure we have addressed all your concerns satisfactorily. If there are any additional points or feedbacks you'd like us to consider, please let us know. Your insights are invaluable to us, and we are eager to address any remaining issues to improve our work.
>
> Thank you for your time and effort in reviewing our paper!

---

### Comment · Area_Chair_Pbh9 · 2025-11-23
**Action Needed: Review Rebuttal and Update Evaluation**

Dear Reviewers,

Thank you, as always, for your valuable contributions and efforts. The authors have now submitted their rebuttal. Please take a moment to review it and provide any necessary follow-up actions, such as additional questions, clarification requests, or updates to your review.

Since the initial ratings ranged from 2 to 6, I kindly ask you to pay close attention to the perspectives of the other reviewers when preparing your final response.

Thank you again for your support.

---

### Meta-Review · Area_Chair_ppRu · 2026-01-06

**Summary:**

This paper presents CudaForge, a training-free agent framework that generates optimized CUDA kernels. The proposed framework makes use of NCU metrics that provide direct feedback from the hardware to help pinpoint the performance bottlenecks. The proposed framework outperforms several baselines on KernelBench and generated a very impressive kernel for cross-entropy loss calculation.

The primary concerns around the work stem from its novelty. The proposed framework is very similar to existing self-refine style frameworks. While there is a potential for novelty if existing frameworks couldn't be applied directly to a new domain, the coder-judge framework used in CudaForge is not meaningfully different than self-refine itself.

The primary novelty of this work relies in the NCU metrics. While simply using the NCU metrics would likely not be enough novelty to warrant publication, there are key design decisions around how those metrics are utilized. These decisions are where the novelty of the work lies. However, the analysis of these decision is unfortunately lacking. There is little analysis as to why these 24 metrics and, in the authors own ablations, it is shown that a set of 20 metrics would perform better. Further, this set of metrics is static. No consideration is given to having the set of metrics be problem specific, kernel specific, or class specific (i.e. memory vs. compute-bound problems).

There are clearly signs of promise in this paper, however, more analysis and experimentation into the use of NCU metrics is needed (and may result in even better performance) for the paper to be recommended for publication.

**Reviewer Concerns:**

## Reviewer wyaT

- Task shapes are too small. This exacerbates the overheads induced by not using torch.compile as the baselines. I believe this concerned was addressed.
- The results on cross-entropy seems too surprising. I do not agree with this concern. Cross-entropy loss has seldom been the compute bottleneck in model training. It is also a operation that is memory bound. The implementation in PyTorch is very general and not overly specific to any given GPU. The implementation generated by CudaForge can be arbitrarily tailed to a given GPU. Given this, a 4x speed-up is not unreasonable.
## Reviewer TYZF

- The overall framework is quite similar to existing self-refine works. I believe this concern is still outstanding.

## Reviewer PHnN

- The two-agent workflow is similar to prior work. While the work the reviewer cited is consider concurrent by ICLR policies, the sentiment remains true.
- Incorporation of NCU metrics is not novel. I am not sure exactly what the reviewer meant by this concern and thus am disregarding it.
- The paper offers limited insight and analysis into its choices of NCU metrics. I believe this concern is still outstanding.

**Reviewer Scores:**

I do not believe reviewer scores would have meaningfully changed.

---

### Decision · Program_Chairs · 2026-01-26

Reject